# Reactive nitrogen in and around the northeastern and Mid-Atlantic US: sources, sinks, and connections with ozone

Min Huang[1,2], Gregory R. Carmichael[3], Kevin W. Bowman[4], Isabelle De Smedt[5], Andreas Colliander[4], Michael H. Cosh[6], Sujay V. Kumar[1], Alex B. Guenther[7], Scott J. Janz[1], Ryan M. Stauffer[1], Anne M. Thompson[1], Niko M. Fedkin[1], Robert J. Swap[1], John D. Bolten[1], Alicia T. Joseph[1]

[1]Earth Sciences Division, NASA Goddard Space Flight Center, Greenbelt, MD 20771, USA
[2]Earth System Science Interdisciplinary Center, University of Maryland, College Park, MD 20740, USA
[3]College of Engineering, University of Iowa, Iowa City, IA 52242, USA
[4]Jet Propulsion Laboratory, California Institute of Technology, Pasadena, CA 91109, USA
[5]Royal Belgian Institute for Space Aeronomy, 1180 Brussels, Belgium
[6]Hydrology and Remote Sensing Laboratory, US Department of Agriculture, Beltsville, MD 20705, USA
[7]Department of Earth System Science, University of California at Irvine, Irvine, CA 92697, USA

*Correspondence to*: Min Huang (minhuang@umd.edu)

**Abstract.** This study describes an application of a regional Earth system model (NASA-Unified Weather Research and Forecasting with online chemistry) with updated parameterizations for selected land-atmosphere exchange processes and multi-platform, multidisciplinary observations. First, we estimate reactive nitrogen (Nr = oxidized $NO_y$ + reduced $NH_x$) emissions from anthropogenic and natural sources, nitrogen dioxide ($NO_2$) column densities and surface concentrations, total and speciated Nr dry or/and wet deposition fluxes during 2018–2023 over the northeastern and Mid-Atlantic US, most of which belong to nitrogen oxides-limited or transitional chemical regimes. The estimated multi-year Nr concentrations and deposition fluxes are then related to ozone ($O_3$), in terms of their spatiotemporal variability and key drivers as well as possible ecosystem impacts. Finally, through three sets of case studies, we identify and discuss about 1) the capability of land data assimilation (DA) to reduce the uncertainty in modeled land surface states at daily-to-interannual timescales that can propagate into atmospheric chemistry fields; 2) the impacts of irrigation on land surface and atmospheric fields as well as pollutants' ecosystem uptake and impacts; and 3) the impacts of transboundary air pollution during selected extreme events on pollutants' budgets and ecosystem impacts. With the updated model parameterizations and anthropogenic emission inputs, the eastern US surface $O_3$ modeled by this tool persistently agrees better with observations (i.e., with root-mean-square errors staying within 4–7 ppbv for the individual years' May-June-July) than many of those in literature where model errors are often tens of ppbv. The model-based correlation between daytime surface $O_3$ and early afternoon $NO_2$ columns, which shows a dependency on column $HCHO/NO_2$ ratios, is higher in 2020 ($r$=0.62) than in other years during 2018–2023 ($r$=0.47–0.56). The $O_3$ vegetative uptake overall dropped by ~10% from 2018 to 2023, displaying clearer downward temporal changes than the total Nr deposition due to the declining $NO_y$ emission and deposition fluxes competing with the increasing $NH_x$ fluxes. It is highlighted that, temporal variability of Nr and $O_3$ concentrations and fluxes on subregional-to-local scales respond to hydrological variability that can be influenced by precipitation and controllable human activities such

as irrigation. Deposition processes and biogenic emissions that are highly sensitive to interconnected environmental and plants' physiological conditions, as well as extra-regional sources (e.g., $O_3$-rich stratospheric air and dense wildfire plumes from upwind regions), have been playing increasingly important roles in controlling pollutants' budgets in this area as local emissions go down owing to effective emission regulations and COVID lockdowns. To better inform the design of mitigation and adaptation strategies, it is recommended to continue evaluating and improving the model parameterizations and inputs relevant to these processes in seamlessly coupled multiscale Earth system models using laboratory and field experiments in combination with satellite DA which would in turn benefit remote sensing communities.

## 1 Background, motivation, and goals

Nitrogen oxides ($NO_x$) are an important group of ozone ($O_3$) precursor and destroyer, and ground-level $O_3$ is a US Environmental Protection Agency (EPA)-regulated criteria air pollutant. $NO_x$ consists of nitric oxide (NO) and nitrogen dioxide ($NO_2$), the latter of which is another US EPA-regulated criteria air pollutant that has the highest exposure disparities (Liu et al., 2021). Emitted from various anthropogenic (anth) and natural sources, $NO_x$ is readily transformable to/from other forms of reactive nitrogen (Nr = oxidized $NO_y$ + reduced $NH_x$) species, such as ammonia ($NH_3$), peroxyacetyl nitrate (PAN) and nitric acid ($HNO_3$). Some of these chemical reactions also contribute to fine particulate matter pollution, which is connected with $O_3$ via aerosol radiative effects and heterogeneous chemistry (Seinfeld and Pandis, 2016; Monks et al., 2021). Many previous studies have demonstrated that $NO_x$ emissions and concentrations play more crucial roles than volatile organic compounds (VOCs) in regulating the magnitude and spatiotemporal variability of $O_3$ (e.g., Duncan et al., 2010; Jin et al., 2017; Koplitz et al., 2022; Souri et al., 2023) as well as aerosols (Carlton et al., 2010; Holt et al., 2015) in much of the northeastern and Mid-Atlantic states, the most populous US region where the land surface is highly heterogeneous and hydroclimatic extremes and exceedances of the US National Ambient Air Quality Standards occur from time to time (US Global Change Research Program, 2023; US EPA, 2023). An improved understanding of the sources, sinks, and distributions of $NO_x$ and Nr as well as how these have been and will be changing through time is beneficial for interpreting $O_3$ air pollution levels and their spatiotemporal variability in this area. The removals of Nr, $O_3$, and other chemicals involved in their life cycles from the atmosphere through wet or/and dry deposition closely interact with multiple other interconnected environmental stressors (e.g., temperature, humidity, precipitation, soil moisture, SM, and carbon dioxide, $CO_2$) and plants' physiological conditions. Together, they can cause intertwined and cascading effects on the diverse terrestrial and aquatic ecosystems (e.g., United Nations Economic Commission for Europe, 1999; Galloway et al., 2003, 2004; Felzer et al., 2009; Simpson et al., 2014; Lombardozzi et al., 2015; Mills et al., 2018; Walker et al., 2019; Clifton et al., 2020; Emberson, 2020) in this area. Due to effective environmental regulations and unusual situations such as COVID lockdowns, anth emissions continue to decrease there. For studies on Nr and $O_3$, attention should also be given to quantifying the impacts of multiple climatic factors as well as nonlocal air pollution sources such as those imported from upwind US regions, Canada, and the

stratosphere, which are partially controlled by the Bermuda High and other pressure systems (e.g., Colarco et al., 2004; Zhu et al., 2013; Ott et al., 2016; Rogers et al., 2020).

Previous global and regional modeling studies have shown that reproducing the observed warm-season Nr and surface $O_3$ levels in the US East is challenging (e.g., Fiore et al., 2009; Chai et al., 2007, 2013; Lapina et al., 2014; Huang et al., 2017a; Lin et al., 2017). The estimated background $O_3$ therein, as well as the importance of its individual contributors, varies substantially among models. Often, the large model-observation mismatches in surface $O_3$ of up to tens of ppbv were not well explained or attributed mainly to the models' uncertain/outdated anth emission inputs. Some of these studies implemented advanced chemical data assimilation (DA) methods to reduce the errors in their predicted surface $O_3$ states by ~50% (Chai et al., 2007). They did not improve the mechanistic representations of $O_3$ related processes, which are of higher policy-relevance and would lengthen the impacts of chemical DA since the model initializations/analysis times. The large uncertainty in model results limits our capability of understanding air quality over these regions and evaluating potential strategies to mitigate the air pollution impacts. High-resolution Earth system modeling with proper model parameterizations, up-to-date inputs, and comprehensive, process-based analysis aided by cross-disciplinary observations can help elucidate the various factors controlling Nr and $O_3$ (Fig. 1a) to better assist with assessing their environmental impacts from past to future.

This study is designed to support the International Global Atmospheric Chemistry-Tropospheric Ozone Assessment Report (TOAR) Phase II activity, which aims to further examine the distributions, temporal changes, and impacts of $O_3$ and its key precursors. A regional Earth system model is applied with updated parameterizations for selected land-atmosphere exchange processes (Section 2.1), running over the Northeast and Mid-Atlantic states for multiple years at 10 km horizontal resolution that is considered to be able to better capture $NO_x$ lifetime and budgets than coarser resolution systems (Li et al., 2023). The model is used together with multiplatform, multidisciplinary observations (Section 2.2) and a range of analysis methods (e.g., model evaluation and diagnosis, formal DA, and sensitivity simulations, Section 2.3) to help achieve the following specific goals: 1) to estimate Nr emissions from various anth and natural (e.g., soil NO and nitrous acid, HONO) sources, $NO_2$ surface concentrations and column densities, total and speciated Nr dry or/and wet deposition fluxes during 2018–2023, with discussions on key anth and environmental/climatic drivers of their spatiotemporal variability during this period (Section 3.1); 2) to relate Nr and $O_3$ concentrations as well as their deposition fluxes during 2018–2023, in terms of spatiotemporal variability, reactions to environmental and biophysical stresses, and potential ecosystem impacts (Section 3.2); and 3) through three sets of case studies (Section 3.3), to discuss in detail about land-atmosphere exchange processes which have been understudied topics. Specifically, we demonstrate the capability of land DA to reduce the uncertainty in the modeled land surface states, land-atmosphere exchange processes and atmospheric states at daily-to-interannual timescales; the impacts of controllable human activities such as irrigation on land surface and atmospheric fields and pollutants' ecosystem uptake; and the impacts of transboundary air pollution during selected extreme events on air pollutants' budgets

and ecosystem impacts. These case studies also help identify sources of model uncertainty before we draw conclusions and outline future directions for further advancements in related areas in Section 4.

## 2 Methods

### 2.1 Coupled modeling system and the baseline simulation

On a 10 km, 63 vertical layer Lambert conformal grid (Fig. 1b–c) from the subsurface to ~100 hPa, the NASA-Unified Weather Research and Forecasting model with online chemistry (WRF-Chem) simulations were conducted over the Northeast and Mid-Atlantic states for 2018–2023 growing seasons starting from 25 April of each year. The analysis of the baseline simulation was focused on May-June-July (MJJ) of 2018–2020, 2022, and 2023. MJJ falls within the plant growing and $O_3$ seasons when atmospheric Nr and $O_3$ most actively interact with ecosystems (Li et al., 2015; Clifton et al., 2020). Year of 2021 is not an emphasis in this paper partly due to the lack of reliable information to represent the COVID impacts on anth emissions for that year. The four-layer Noah-Multiparameterization (MP, Niu et al., 2011) land surface model (LSM) version 3.6 within the NASA Land Information System served as the land component of this modeling system, running with a sprinkler irrigation and the Community Land Model type of SM factor controlling stomatal resistance (i.e., $\beta$ factor) schemes. Noah-MP was forced by the North American Land Data Assimilation System Phase 2 forcing data during the long-term (since 2000) offline spin-up. Noah-MP's $CO_2$ forcings for 2018, 2019, 2020, 2022, 2023's warm seasons were set to 410, 412, 415, 420, and 423 ppmv, respectively, based on measurements at the Mauna Loa Observatory and its nearby Maunakea Observatories for part of 2023 (https://gml.noaa.gov/webdata/ccgg/trends/co2/co2_mm_mlo.txt, last access: 12 January 2024). This advanced Noah-MP's default setup in terms of appropriately representing the global $CO_2$ growth rates of 2–3 ppmv year$^{-1}$ for recent years (https://gml.noaa.gov/ccgg/trends/gl_gr.html, last access: 12 July 2024). Ignoring the spatial and (intra)seasonal variability in $CO_2$ of up to tens of ppmv over the study region (Karion et al., 2020) may have introduced only small uncertainty in photosynthesis and deposition modeling according to independent model sensitivity analysis in which $CO_2$ forcings were perturbed (e.g., Sun et al., 2022). The land use/land cover (LULC) and soil type inputs of Noah-MP were based on the 20-category International Geosphere-Biosphere Programme-modified Moderate Resolution Imaging Spectroradiometer (MODIS, Fig. 1b) and the 16-category State Soil Geographic (Fig. S1) datasets, respectively. Crop-type and irrigation map/fraction information required by the irrigation scheme came from Monfreda et al. (2008) and Salmon et al. (2015), respectively, the latter of which (Fig. 1c) incorporated MODIS information.

Major atmospheric and land model physics as well as chemistry schemes were configured in similar ways to those in Huang et al. (2022). The photosynthesis-based dry deposition approach recommended in Huang et al. (2022) and a number of other previous dry deposition studies cited therein was applied to most gaseous species. No change was made to sulfur dioxide dry deposition approach (Erisman et al., 1994) for this study. The modeled wet deposition fluxes were also evaluated and discussed in this work. In replacement of the metric-based approach in Huang et al. (2022), $O_3$ vegetative impacts were

dynamically modeled by applying two separate factors to photosynthesis and stomatal conductance rates (Lombardozzi et al., 2015) that are calculated in Noah-MP. These factors are land cover-dependent functions of $O_3$ uptake cumulated during growing season when leaf area index (LAI) exceeds 0.5. To account for the ability of plants to detoxify $O_3$, $O_3$ fluxes were only accumulated when they exceeded a threshold of 1.0 nmol $O_3$ $m^{-2}$ $s^{-1}$. As demonstrated in previous offline (Lombardozzi et al., 2015) and online (Li et al., 2016; Sadiq et al., 2017) modeling studies, dynamically modeling $O_3$ vegetative impacts could help quantify the perturbations of $O_3$ to a variety of hydrological, ecological, and weather variables. Online-calculated

biogenic emissions of $O_3$ precursors such as VOCs and Nr species in the simulations were adjusted to be more sensitive to multiple environmental stresses. Specifically, a drought adjusting factor $\gamma_d$ was introduced in the Model of Emissions of Gases and Aerosols from Nature (MEGAN) biogenic isoprene emission calculations following the suggestions by Jiang et al. (2018), which depends on the $\beta$ factor and the maximum carboxylation rate. The plant function type information needed for

MEGAN was converted from the annual European Space Agency Climate Change Initiative (ESA CCI) land cover product for 2018–2020, and the 2020 data from this product were also used for the years afterwards. The Noah-MP modeled LAIv (i.e., LAI/Green Vegetation Fraction) feeds into MEGAN calculations. Soil emissions of NO were estimated largely based on the mechanism recommended by Hudman et al. (2012) and Simpson and Darras (2021), i.e., for dry and wet soils that are determined by a SM index (i.e., a function of SM, soil wilting point and field capacity), different sets of biome-based

emission coefficients (Steinkamp and Lawrence, 2011) and the standing Nr pool plus nitrogen input from deposition being adjusted by water-filled pore space $\theta$ (i.e., SM divided by porosity), soil temperature (Wang et al., 2021), and canopy reduction factor. The pulsing effects, which are small for this study area/season, were accounted for. Soil HONO emissions were also calculated online, scaled from soil NO emissions using biome-dependent factors specified in Table A1 of Rasool et al. (2019) that were partly adapted from Oswald et al. (2013). Nitrogen input from fertilizer was not included in the soil

emissions calculations to avoid double counting with agricultural emissions from the anth emission input to be introduced below. Oceanic natural $NH_3$ emissions were not included, which were estimated to have negligible impacts on Nr overland (Paulot et al., 2013). Lightning emissions were also calculated online and vertically distributed adopting the setup described in Huang et al. (2021) which was based on cloud-top-height-based parameterizations (Wong et al., 2013) and climatological intra-cloud to cloud-to-ground flash ratios. A passive lightning $NO_x$ tracer was again implemented, that experienced

atmospheric transport but not chemical reactions. Aerosol direct, semidirect and indirect radiative effects were enabled.

    Emissions from various anth source sectors came from the Copernicus Atmosphere Monitoring Service (CAMS) global inventory version 5.3, available at 0.1°×0.1° horizontal resolution with monthly and year-by-year variability. To account for COVID impacts, for 2020, grid- and sector-dependent factors (Doumbia et al., 2021) were applied to adjust the emissions.

This CAMS inventory for the years after 2015 was developed by extrapolating the Emissions Database for Global Atmospheric Research version 5 based on the Community Emissions Data System version 2 trends and including emissions from ships as well as monthly variability that were estimated separately (Granier et al., 2019; Soulie et al., 2024). It's noted in Elguindi et al. (2020) and references therein that $NO_x$ emissions for recent decades from an earlier version of CAMS

inventory do not notably differ from other bottom-up inventories over the US where more detailed information for emission inventory developments are available. In contrast, top-down estimates diverge significantly due to uncertainty in the used satellite $NO_2$ retrievals as well as the model representations of various atmospheric processes many of which are scale-dependent. A clear understanding of the impact of background $NO_x$ sources, including natural emissions, on constraining $NO_x$ emissions with satellite $NO_2$ data is urgently needed. The 0.1°×0.1°, version 2.6r1 of the Quick Fire Emissions Dataset (QFED, Darmenov and da Silva, 2015), developed with the fire radiative power approach, was applied with plume rise (Grell et al., 2011). QFED $NO_x$ emissions over North America during 2012–2019 are in comparable magnitudes with other widely-used fire emission datasets such as the Fire INventory from NCAR (FINN) while its $NH_3$ emissions are higher than the estimates from other products (Wiedinmyer et al., 2023). Figure 2 presents the total anth and biomass burning (fire) $NO_x$ and $NH_3$ emissions averaged for each year's MJJ. Anth $NO_x$ emissions are shown to decrease due to effective emission controls (i.e., a -16.3% overall change from 2018 to 2023), except for slight increases along a few shipping lanes. They are anomalously low in 2020 (~23.8% lower than 2018) largely due to reduced human activities during the COVID lockdowns. The temporal changes in non-methane (NM) VOC emissions are relatively smaller, with the domain-mean in 2023 only ~6% lower than in 2018. The total anth $NH_3$ emissions were growing in many places, most evidently over croplands as a result of the rising agricultural soil and livestock emissions. The QFED-based fire $NO_x$ and $NH_3$ emissions were generally increasing, reaching their highest in 2023.

Daily reinitialized atmospheric initial conditions (ICs) and boundary conditions (BCs) were downscaled from the 3-hourly, 32 km North American Regional Reanalysis (NARR) dataset, which overall well represents the observed daily variability in apparent temperature for the eastern US (e.g., Ibebuchi et al., 2024). Huang et al. (2017b) showed that, initializing WRF with the North American Mesoscale Forecast System (6-hourly, 12 km)'s atmospheric fields instead of NARR's did not result in significant changes in WRF-simulated surface air temperature fields over the southeastern US. A set of the 6-hourly Community Atmosphere Model with Chemistry (CAM-Chem, for 2018–2020, 0.9°×1.25°/56 vertical levels) and Whole Atmosphere Community Climate Model (WACCM, beyond 2020, 0.9°×1.25°/88 vertical levels) simulations that also ingested QFED fire information served as the chemical BCs of the WRF-Chem baseline simulation because of its higher completeness of chemical species, and for WACCM, its availability for very recent years compared to chemical reanalysis products which are likely to be more accurate. The chemical BC models' stratospheric $O_3$ tracer fields also supported our multi-year analysis and a case study (Section 3.3.3). From 2018 to 2023, the lower free tropospheric $O_3$ in MJJ first rose by up to 4 ppbv, and then dipped down by up to 4–6 ppbv before rising again (Fig. 3a). The interannual variability in lower free tropospheric $O_3$ and its precursors upwind of the eastern US, as well as the synoptic wind fields that shifted from westerly in 2018–2022 to northwesterly in 2023 (Fig. 3c), play critical roles in controlling the modeled large-scale $O_3$ patterns and their temporal changes. Ozone transport from the stratosphere more strongly influenced the lower free tropospheric $O_3$ over the southern part of our domain in 2023 than in 2018 by up to 4 ppbv (Fig. 3a–b). Although the stratospheric air influences on

surface $O_3$ were diluted to no more than a few ppbv (Fig. S2), the challenges regional models experience in reproducing their magnitudes and interannual variability may introduce uncertainty to the estimated surface $O_3$ changes.

## 2.2 Observations

### 2.2.1 Chemical observations from satellites, aircraft, and ozonesondes

The TROPOspheric Monitoring Instrument (TROPOMI) on board the Copernicus Sentinel-5 Precursor satellite launched in 2017 provides trace gas and aerosol measurements at daily global coverage since April 2018, with ascending node ~13:30 local time overpasses. It has much finer resolutions (i.e., $3.5\times5.5$ km$^2$ at nadir since August 2019, and $3.5\times7$ km$^2$ before then), a wider spectral range and higher signal-to-noise ratio per ground pixel than its predecessors. TROPOMI data have

demonstrated their robustness in studying air pollution from numerous source sectors (e.g., land and water traffic, power plants, oil, gas and other industries, biogenic and fire) in greater detail (e.g., Georgoulias et al., 2020; van der Velde et al., 2021; Griffin et al., 2021; Goldberg et al., 2021; Dix et al., 2022). In this study, the gridded ($0.02°\times0.02°$) monthly and daily level 2 TROPOMI tropospheric vertical column $NO_2$ data were analyzed together with WRF-Chem fields to help understand the temporal changes in column $NO_2$. The gridded ($0.05°\times0.05°$) monthly TROPOMI formaldehyde (HCHO) tropospheric

vertical columns (De Smedt et al., 2021) were also used to calculate HCHO/$NO_2$ ratios to help determine $O_3$ chemical regimes over the study area. The TROPOMI-based HCHO/$NO_2$ ratios were supplemented by those derived from the gridded (1 km$\times$1 km) $NO_2$ and HCHO data collected on selected days of 2018 over New York City and the Long Island Sound by two similar airborne instruments (Judd et al., 2020) Geostationary Trace gas and Aerosol Sensor Optimization (GeoTASO) and GEO-CAPE Airborne Simulator (GCAS).


Additionally, to help identify and attribute air pollutants during highly polluted events in 2023 (Section 3.3.3), the Joint Polar Satellite System-1 Cross-track Infrared Sounder (JPSS-1/CrIS, with descending/ascending nodes of ~1:30/13:30 local time) $O_3$, carbon monoxide (CO), and PAN level 2 daily summary data provided by the TRopospheric Ozone and Precursors from Earth System Sounding project, were analyzed. The analysis of these extreme events was also supported by eight

ozonesondes launched from the Virginia Commonwealth University Rice Rivers Center (RRC, 37.33197°N, 77.20842°W) during the inaugural edition of NASA Student airborne Research Program (SARP)-East campaign in summer 2023 along with model results and ground-based observations (Section 2.2.3).

### 2.2.2 Satellite SM and precipitation products

To characterize drought conditions and their temporal variability, which interact with atmospheric chemistry, NASA's L-band Soil Moisture Active Passive (SMAP) 9 km enhanced surface (first 5 cm belowground) SM (SSM) data version 5 were utilized, as well as the version 7 of daily precipitation data from the NASA-JAXA Global Precipitation Measurement (GPM) produced at $0.1°\times0.1°$ resolution using the Integrated Multi-satellitE Retrievals for GPM-Final run algorithm. Despite the different sampling strategies and retrieval algorithms of SMAP and GPM, interannual variability in the drought conditions

indicated by these SSM and precipitation data are qualitatively consistent (Fig. 4), which are also consistent with information from independent sources such as the North American Drought Monitor (https://droughtmonitor.unl.edu/NADM, last access: 12 July 2024; see Table S1 and Fig. S3 for further analysis and discussions). In addition to rainfall, irrigation water and other elements relevant to water and energy balances can also impact the variability in SSM which has feedback to the regional precipitation patterns. The wide range of the SMAP SSM from <0.2 to >0.5 $m^3$ $m^{-3}$ and its interannual differences which

often exceed 0.1 $m^3$ $m^{-3}$, indicate the diverse SM regimes (i.e., dry, transitional, and wet) and therefore spatially and temporally varying land-atmosphere coupling strengths (Seneviratne et al., 2010, and references therein). The varying SSM-temperature coupling strengths were determined based on WRF-Chem results, with support of the 0.25°×0.25° European Centre for Medium-Range Weather Forecasts Reanalysis version 5 (ERA5) surface air temperature field. In a case study (Section 3.3.1), SMAP SSM data were assimilated into the Noah-MP LSM to improve the land ICs of WRF-Chem, and

further, the modeled weather and atmospheric chemistry fields.

### 2.2.3 Ground-based observations

Hourly surface ultraviolet absorbance $O_3$ observations from the US EPA's Air Quality System (AQS, a major source of the TOAR database, last update in August 2024) were used to support the quantification of $O_3$ temporal variability and model

evaluation. The AQS $NO_2$ observations, which have poorer spatial coverage than their $O_3$ data, were also examined to help qualitatively understand surface $NO_2$ variability. These AQS $NO_2$ measurements made using the chemiluminescence detection with catalytic conversion are known to be positively biased by up to 50% due to $NO_z$ ($NO_y$-$NO_x$) interferences (e.g., Dunlea et al., 2007). Speciated aerosol measurements offered by the Clean Air Status and Trends Network (CASTNET) and AQS, CASTNET $HNO_3$, the National Atmospheric Deposition Program (NADP)/Ammonia Monitoring

Network (AMoN) $NH_3$ observations, as well as the NADP/National Trends Network (NTN) wet deposition fluxes and precipitation data, were used to infer or directly evaluate WRF-Chem's deposition performance. Deposition datasets from other studies, some of which integrated surface or/and satellite observations with other models (e.g., Schwede and Lear, 2014; Fu et al., 2022; Rubin et al., 2023), will be referred to in the discussions.

Additional datasets for selected time periods were used in DA case studies to help interpret and validate the model results (Section 3.3.1). These include gauge-based precipitation data and SSM measured using HydraProbe sensors at Harvard Forest, Massachusetts (42.53523°N, 72.17393°W) and a US Climate Reference Network (CRN) site in Millbrook, New York (41.786°N, 73.74°W) during the July 2022 SMAP validation experiment (SMAPVEX22); and surface air temperature observations in July 2018 and 2022 from the National Centers for Environmental Prediction (NCEP) Global Surface

Observational Weather Data product.

## 2.3 Case studies and sensitivity simulations

Temporal variability of Nr and $O_3$ concentrations and fluxes at subregional-to-local scale are partially driven by hydrological variability which can be influenced by both precipitation and human activities such as irrigation. Two sets of modeling and DA case studies (Sections 2.3.1 and 2.3.2) were conducted to show that the modeled land surface states, such as SM, can be improved via land DA and/or updating the model's irrigation schemes, which further impacts the modeled land-atmosphere exchange processes and atmospheric fields.

### 2.3.1 Effects of SM DA on modeled $NO_2$ and $O_3$

For this case study, SMAP morning-time (~6 am local time) SSM data were bias-corrected via matching the means and standard deviations of SMAP and Noah-MP SSM monthly climatology. The bias-corrected data were then assimilated into the Noah-MP LSM using a 40-member ensemble Kalman filter approach to adjust WRF-Chem's land ICs during July 2018 and July 2022. Meteorological forcing (precipitation, short- and longwave radiation) and state (Noah-MP SM) perturbation attributes were set up largely based on Kumar et al. (2009) recommendations for the Noah LSM, and the input observation error standard deviation was set to be $0.04\,\mathrm{m^3\,m^{-3}}$ according to the SMAP data quality requirement. Through this experiment we evaluate whether and to what extent can satellite SM DA improve the day-to-day (i.e., before and after a precipitating event during the SMAPVEX22 campaign when in situ SSM data were also collected near the SMAP morning overpassing times) and interannual variability (i.e., July 2018 and July 2022) of Noah-MP SM, even in dense vegetation regions such as the eastern US where satellite SM retrieval is generally more challenging. How the adjustments to Noah-MP land surface states by DA impacted the modeled atmospheric fields was also quantified.

### 2.3.2 Irrigation impacts on $O_3$ vegetation uptake and Nr deposition

Using flux-based $O_3$ metrics derived from model outputs, recent studies (Mills et al., 2018; Huang et al., 2022) estimated that the negative impacts of ground-level $O_3$ on crop yields are particularly large in humid irrigated and rainfed agricultural lands, where the plants' stomatal uptake of $O_3$ is significant. The global-scale coarse-resolution analysis for 2010–2012 by Mills et al. (2018), which was based on $O_3$-flux metrics, also estimated that irrigation promotes the $O_3$ impacts on wheat production by up to ~10%. To dynamically evaluate in detail the irrigation impacts on land surface and atmospheric fields as well as the estimated $O_3$ and Nr ecosystem impacts across our study area for recent years, WRF-Chem simulations were conducted with three sets of irrigation configurations, defined as (a, b, c) below, and for each of these three scenarios, two simulations were conducted with and without $O_3$ vegetation impacts:

a) Full irrigation (baseline): Sprinkler irrigation occurs in the morning when rootzone SM drops below 50% of field capacity;

b) Reduced irrigation: Sprinkler irrigation occurs in the morning when rootzone SM drops below 25% of field capacity, and the estimated irrigation water usage for this scenario is ~1/6 of the full-irrigation scenario for irrigated areas south of ~37°N in our domain;

and c) Irrigation option was completely disabled.

This sensitivity analysis is focused on 21–30 June 2022, when irrigated fields in the Carolinas that grow mostly $O_3$-sensitive crops were under stress according to the Vegetation Drought Response Index produced by the National Drought Mitigation Center (Fig. S4). This region also encompasses Nr deposition hotpots that have been experiencing critical load (CL) exceedances (i.e., the amount of Nr deposition exceeds the CL threshold, the point above which deposition could harm sensitive ecosystems). For this period, irrigation water consumption under the full-irrigation scenario may be higher than normal, and the estimated surface fluxes under reduced- and no-irrigation scenarios may be particularly smaller than usual and more strongly constrained by SM.

### 2.3.3 Impacts of transboundary pollution on weather, air quality and ecosystems

The Northeast and Mid-Atlantic US air quality is regularly affected by pollutants emitted or/and formed in upwind US states. Actions have been taken to tackle cross-state air pollution such as using the Cross-State Air Pollution Rule framework (https://www.epa.gov/Cross-State-Air-Pollution/overview-cross-state-air-pollution-rule-csapr, last access: 12 January 2024). However, with the US EPA's "Good Neighbor Plan" being put on hold by the Supreme Court (https://www.epa.gov/Cross-State-Air-Pollution/good-neighbor-plan-2015-ozone-naaqs, last access: 12 July 2024), downwind US states may continue to face difficulties in complying with the 2015 $O_3$ National Ambient Air Quality Standards due to the upwind states' pollution impacts. Periodically, distant sources including Canadian wildfires and $O_3$-rich stratospheric air also travel to the northeastern and Atlantic states. Satellite and in situ observations are powerful in detecting such episodic events that occur more frequently in recent years, assisting with early warnings and early actions. To help quantify the impacts of such extreme events, as well as other upwind air pollution sources, on weather, air quality and ecosystems during 13–16 June 2023, two WRF-Chem sensitivity simulations were conducted and analyzed together with the baseline simulation and multiplatform observations. Clean chemical BCs were applied in one of these sensitivity simulations, and WACCM running with the FINN fire emission input served as the chemical BCs of the other WRF-Chem sensitivity simulation named "Sen". Fire emission is identified as one of the most important configurations in global wildfire modeling (e.g., Veira et al., 2015).

### 3 Results and discussions

### 3.1 Nr emissions, concentrations, and deposition fluxes during 2018–2023

The modeled soil NO and HONO emissions vary strongly with SM as well as soil temperature that can be impacted by SM. Even without land DA the model fairly well reproduced the large-scale spatial gradients and interannual variability of soil wetness (Figs. 4a and S5). Soil emissions exhibit notable monthly variations, with multi-year June- and July-mean values ~11% and ~59% higher than the May-mean, respectively, associated with overall warmer and drier conditions. These monthly variations, together with the ~8% and ~18% multiyear June-May and July-May mean differences in anth+fire emissions as well as modeled surface and column $NO_2$ fields, help interpret the higher TROPOMI and AQS $NO_2$ on warmer months over many rural areas, especially those near high-temperature agricultural regions (Fig. S6), a point Goldberg et al. (2021) also highlighted. The maxima and minima of MJJ soil emissions are shown in 2020 and 2018, respectively, and the

interannual variability of soil emissions roughly anti-correlates with that of SM, with correlation coefficient $r$ ranging from -0.63 to -0.40 ($p \ll 0.01$; Fig. 5a;c). For most years, the estimated MJJ-mean soil NO and HONO emissions are particularly high in warm and/or dry areas including parts of the Carolinas, Virginia, New York, Michigan, and Canada's Ontario, where their contributions to the total soil+anth+fire $NO_y$ emissions persistently exceeded 30% (Fig. 5a–b). Based on a global atmospheric chemistry model with a similar soil emission scheme, previous estimates of the soil NO emission contributions to column $NO_2$ for this area were minor compared to other US regions in 2005 (i.e., <15% uniformly, Vinken et al., 2014), when anth $NO_x$ emissions were >25% higher than in 2018 according to the CAMS inventory and other estimates. Owing to the overall declining US anth emissions and the changing climate, soil emissions play an increasingly important role in controlling Nr, and further, $O_3$ air quality, in this area. Accordingly, the needs to properly parameterize soil emissions and accurately model soil environments (e.g., SM, soil temperature, pH) have been growing stronger which could greatly benefit from laboratory and field experiments.

Despite the increasing anth $NH_3$ and fire Nr emission trends (Section 2.1) and the abovementioned interannual variability in soil NO and HONO emissions, the total Nr emissions as well as surface $NO_y$ emissions that contributed to >50% of the total Nr emissions show decreasing year-to-year changes during 2018–2023 except for the dip in 2020 that is mainly attributable to the COVID lockdowns (Fig. 6a). Closely linked to such temporal changes in $NO_y$ emissions, that in many areas overwhelm the effects of slower $NO_2$ and $NO_y$ dry deposition (Fig. S7 and later discussions), the modeled column and surface $NO_2$ both display downward changes since 2018, with their lowest values occurring in 2020 (Figs. 6b and 7). From 2018 to 2023, on average, column and surface $NO_2$ dropped by 15–20%. Impacted by the decreasing $NO_2$, HCHO columns overall demonstrate a few percent slower year-to-year changes than $NO_2$ (Fig. S8), in large part because of less significant NMVOC emission changes (Section 2.1). Impacted mostly by shipping and lightning emissions as well as North American pollution outflows, the amount of $NO_2$ above the ocean is lower than overland. Early afternoon (19 UTC, local standard time +5 or +6, near TROPOMI overpassing times) surface and column $NO_2$ are ~44% and ~29% lower than their daytime averages (13–24 UTC, roughly the sampling times of geostationary missions such as the Tropospheric Emissions: Monitoring of Pollution, TEMPO, and the anticipated Geostationary Extended Observations). The stronger subdaily variability in surface $NO_2$ than in column $NO_2$ reflects the impacts of photochemistry and evolution of planetary boundary layer on the rapidly changing vertical distributions of chemicals throughout the daytime which have also been demonstrated in Huang et al. (2017b) and other studies with aircraft observations. Dependent strongly on convection, lightning NO emissions show high variations from year to year in terms of locations and magnitudes, having larger impacts on free-tropospheric and column-average $NO_2$ than surface $NO_2$ (Fig. S9).

The column $NO_2$ spatiotemporal variability based on WRF-Chem and TROPOMI greatly resemble one another (Fig. 7a–b), and larger model-TROPOMI discrepancies are seen over the areas possibly influenced by lightning NO emissions and transboundary pollution where both model and retrieval errors may be large. The interannual variations in such pollutant

sources aloft may also explain the different interannual variability in surface and column $NO_2$ for some locations and years. AQS $NO_2$ data, although sparsely distributed and positively biased, qualitatively confirmed the model-suggested year-to-year changes in surface $NO_2$ (Fig. S10).

Drought conditions, as well as the opposite directions of $NO_y$ and $NH_3$ emission and concentration changes, helped shape the interannual variability in the total Nr deposition fluxes (Fig. 8a). Overland, the modeled Nr wet deposition fluxes often contributed to nearly or lower than 30% of the total Nr deposition. These contributions are smaller than earlier estimates for this area (e.g., <60% in Tan et al., 2018, where wet deposition based on 11 global models was overestimated), due in part to WRF-Chem wet deposition being underestimated referring to the NADP/NTN measurements (Table S2 and Fig. S11). This

underestimation in wet deposition can be attributed to known limitations in the WRF-Chem wet deposition scheme (Ryu and Min, 2022; Yao et al., 2023). The underestimated model precipitation rates and inaccurate model precipitation patterns on event-to-seasonal scales (Figs. 4b, S5, and S11; Section 3.3.1 case study), as well as observation representation errors, also have caused the negative biases in wet deposition fluxes and the positive biases in aerosol concentrations especially for $NH_4$ and $NO_3$ (Figs. S11–S14). Such precipitation biases in WRF have also been reported in previous studies, and they can

indirectly impact dry deposition modeling.

    Dry $NO_y$ deposition fluxes decreased evidently (i.e., by 5–16% overall and >50% in some populated areas) whereas $NH_x$ dry deposition fluxes show up to ±3% of overall interannual variability and rose by >20% over certain agricultural lands (Fig. S15) where $NH_3$ emissions have been climbing up. Due to not applying a bi-directional approach (Zhang et al., 2010;

Massad et al., 2010; Pleim et al., 2019), these $NH_3$ fluxes may be overestimated over source regions by a few percent (Zhu et al., 2015; Liu et al., 2020a), contributing to the model's minor negative $NH_3$ biases relative to the NADP/AMoN data (Table S2 and Fig. S13). Nevertheless, the contrasting directions of change in $NO_y$ and $NH_x$ deposition fluxes as well as the importance of $NH_x$ deposition in total deposition corroborate results from other studies for earlier periods (e.g., Schwede and Lear, 2014; Li et al., 2015; Jia et al., 2016; Geddes and Martin, 2017; Liu et al., 2020b, and references therein). With several

percent of interannual differences in flux partitioning (Fig. 8b–c), in all years' MJJ, $HNO_3$ and $NH_3$ contributions (>35%) dominated in the Nr dry deposition fluxes. NO dry deposition is negligible due to extremely high surface resistance and in figures is combined with $NO_2$ into $NO_x$ fluxes, that contribute to 12–15% of Nr dry deposition fluxes. Unlike most other species, surface resistance of $HNO_3$ is nearly zero, whose dry deposition variability is therefore driven dominantly by aerodynamic resistance and quasi-laminar sublayer resistance and responds differently to drought conditions than the other

Nr species and $O_3$ (Section 3.2). The modeled $HNO_3$ daytime dry deposition velocities over most forested areas fall within 4–8 cm s$^{-1}$, close to the measurements reported in literature for similar land cover types in the eastern US (e.g., Nguyen et al., 2015). These are ~a factor of 10 higher than the dry deposition velocities of $NO_2$ and PAN, similar to the results in Wu et al. (2011) based also on a photosynthesis-based dry deposition model and the flux measurements summarized by Delaria and Cohen (2023).

Many global models have provided their estimates of total and speciated Nr deposition fluxes for previous decades (e.g., Dentener et al., 2006; Paulot et al., 2018; Tan et al., 2018; Rubin et al., 2023). Here, our regional model results present more details which could be beneficial for estimating CL exceedances on relatively smaller spatial scales. They are overall of a lower magnitude, reflecting the impacts of the declining $NO_y$ and Nr emissions which are anticipated to continue into the

coming decades. This may also be attributed to the impact of the changing climate and the model uncertainty relevant to scales, deficits in deposition schemes and inputs as well as uncounted deposition of certain organic Nr species due to our chosen chemistry and aerosol schemes. Possibly also for these reasons, a little over 50% of the surface Nr emissions were estimated to be removed via deposition in this area for all years (Fig. 6a), slightly lower than the estimates in previous modeling studies.


Comparing our WRF-Chem Nr deposition fluxes to the CL thresholds in Simkin et al. (2016) for herbaceous plants that range from 7.4 to 19.6 kg ha$^{-1}$ a$^{-1}$, from 2018 to 2023, the high likelihoods of CL exceedances in Pennsylvania dropped whereas those in parts of North Carolina may have remained high. The Nr deposition fluxes stayed below these CL thresholds over most of the northern forests, a region where primary productivity has been determined to be nitrogen-limited

(Du et al., 2020) and can be highly sensitive to the interannual variability in Nr deposition (Fig. S16). The empirical CL thresholds of >3–8 kg ha$^{-1}$ a$^{-1}$ for the eastern US forests in Pardo et al. (2011) are higher than the modeled Nr deposition fluxes over the forests in New England states and West Virginia whereas for the other forests roughly close to or lower than the modeled. These results help explain the findings in Horn et al. (2018) that tree growth and survival have increasing and flat-to-slightly-decreasing relationships with Nr deposition for New England/West Virginia forests and other eastern US

forests, respectively. For lichen, WRF-Chem suggests that widespread CL exceedances occurred throughout the study period, according to the static CL thresholds of 3.5, 3.1, 1.9, and 1.3 kgN ha$^{-1}$ a$^{-1}$ for total species richness, sensitive species richness, forage lichen abundance, and cyanolichen abundance, respectively (Geiser et al., 2019). Note that these lichen CL thresholds are likely to be conservative for the eastern US as they were derived partially from biased model deposition fields, and further assessments on the uncertainty of these thresholds are necessary.


### 3.2 Spatiotemporal variability of Nr and $O_3$ concentrations and deposition fluxes

The interannual, day-by-day and subdaily variability in HCHO/$NO_2$ ratios derived from TROPOMI and airborne GCAS and GeoTASO data indicates the variable photochemical environments driven by the changing meteorology and emissions, but, as noted in a number of prior studies (e.g., Duncan et al., 2010; Jin et al., 2017; Tao et al., 2022; Souri et al., 2023), can also

be affected by retrieval uncertainty and several other types of errors. Yet, they indicate that, much of the study area belong to $NO_x$-sensitive or transitional chemical regimes during 2018–2023 (i.e., HCHO/$NO_2$ higher than empirical thresholds of 2–4, Fig. 9) except very few megacities such as the Greater New York City and Toronto, Canada, and for those urban regions, $O_3$ formation continues the trends of turning sensitive to $NO_x$.

Largely explainable by the changing $NO_y$ emissions and $NO_x$-sensitive chemical regimes, the spatial patterns of the modeled interannual differences in column $NO_2$ and surface $O_3$ concentrations roughly resemble one another. Both $NO_2$ and $O_3$ display downward changes over the majority of terrestrial areas whereas the opposite direction of changes over the Atlantic Ocean (Figs. 7b and 10a). In more than half of the terrestrial model grids, the interannual variability of 19 UTC $NO_2$ columns and daytime surface $O_3$ are moderately correlated ($r>0.6$), with the $r$ value of 0.57 averaged across all overland

grids and 0.92 for grids where the $p$ values of the correlation tests are lower than 0.05. Fig. 11 indicates the connection between early afternoon (19 UTC) $NO_2$ columns and daytime surface $O_3$ as well as the dependency of this connection on column $HCHO/NO_2$ ratios. Larger-than-two $HCHO/NO_2$ values dominate the study region where the overall surface $O_3$-$NO_2$ column spatial correlation is high ($r=0.54$). Daytime surface $O_3$ concentrations exhibit the most robust spatial correlation with early afternoon $NO_2$ columns in 2020 ($r=0.62$, versus 0.47–0.56 for other years), when the domain-wide median and

mean $HCHO/NO_2$ ratios are larger than the other years' by at least 0.5. These model results suggest that $NO_x$-sensitive or transitional regimes dominate this region during 2018–2023 and point to a potential of inferring surface $O_3$ variability across this area from high-quality remote sensing $NO_2$ and HCHO column data.

The reduction in $NO_y$ emissions contributed to the domain-average changes in median (-0.7 ppbv) and mean (-1.0 ppbv)

daytime surface $O_3$ concentrations overland from 2018 to 2023, which are much smaller than that in $95^{th}$% $O_3$ (by -3.5 ppbv). The lowering $NO_y$ emissions also resulted in less titration, and consequently, the slightly increased $5^{th}$% $O_3$ (by 0.3 ppbv). Such modeled general directions of $O_3$ temporal changes in this area over the past ~5 years are qualitatively consistent with Cooper et al. (2012) for springs and summers of 1990–2010 as well as follow-on studies (Simon et al., 2015; Lin et al., 2017; Gaudel et al., 2018) and the US EPA's periodically-updated $O_3$ trend summary (https://www.epa.gov/air-trends/ozone-

trends, last access: 12 July 2024). The model captured the COVID-induced daytime surface $O_3$ reductions in 2020 (i.e., overland, ~0.8 ppbv lower than in 2019 on average) that have also been reported in numerous independent studies (e.g., Keller et al., 2021; Steinbrecht et al., 2021; Putero et al., 2023). The interannual variability of imported $O_3$ and its precursors from other regions, as well as the interconnected environmental and plant physiological conditions (e.g., via soil-vegetation-atmosphere interactions whose strengths vary in space and time) modulated biogenic VOC emissions, deposition, chemical

reactions, transport and mixing, also drove the $O_3$ changes on regional-to-subregional scales.

The spatial patterns of WRF-Chem modeled surface $O_3$ broadly match the AQS observations for most of the years (Fig. 10), with root-mean-square errors (RMSEs) ranging from 4.0 to 6.5 ppbv which are significantly lower than the magnitudes of tens of ppbv in many earlier modeling studies for the similar regions. The better performance may have substantially

benefited from the advancements in model parameterizations and the updated anth emission inputs. Although WRF-Chem surface and column $NO_2$ temporal changes agree well with the observed, the model struggled to capture the observed deviations of surface $O_3$ in 2023 from previous years, likely due to its failure in representing the particularly strong

influences of stratospheric $O_3$ or/and other extra-regional sources on (near-)surface $O_3$ in 2023 (Figs. 3b and S2). Later in a case study, the dependency of WRF-Chem $O_3$ performance on how well transboundary pollution as well as regional climatic conditions and their driving processes are represented in the model will be investigated further.

Similar to dry deposition of Nr species and conclusions from Huang et al. (2022), the spatiotemporal variability of $O_3$ dry deposition velocities is closely linked with land cover types, environmental and vegetation conditions, with their highest daytime-average values ($v_{d,o3}$ >1.0 cm $s^{-1}$) seen over moist forests and >30% lower daytime-average values over croplands experiencing drier conditions (Figs. S7 and S15). Cumulative stomatal $O_3$ uptake (CUO), a recommended metric for assessing the potential $O_3$ vegetation impact, that is affected by stomatal conductance, boundary layer resistance, and surface $O_3$ levels, appears also high over the croplands in Ohio and Indiana (~40 mmol $m^{-2}$) where surface $O_3$ concentrations are high while much lower over drier croplands in the Carolinas (<30 mmol $m^{-2}$). Except for regions influenced by the wetter-than-normal conditions or/and increasing surface $O_3$ concentrations, the CUO fields show declining trends (i.e., overall dropped by ~10% from 2018 to 2023). Our results are qualitatively consistent with those in Clifton et al. (2020) for the northeastern US, where based on a global model, stomatal $O_3$ uptake cumulated through MJJ 2010 with no detoxification threshold was estimated to be ~35 mmol $m^{-2}$. Their modeled flux was projected to decrease under the Representative Concentration Pathways 8.5 future scenario under which soil may be drier than present day conditions over the eastern US (Cook et al., 2020). As indicated in Fig. 12a, our modeled CUO values are higher over croplands and forests than shrub/grass averagely and more spatially variable. These CUO fluxes display clearer trends in most grids than the total Nr deposition fluxes, due to $NO_y$ and $NH_x$ deposition fluxes having competing directions of changes through the past years (Figs. 8b–c and S15). The potential impacts of Nr deposition are strongest and weakest on croplands and water, respectively (Fig. 12b).

## 3.3 Three case studies

### 3.3.1 Land DA

Satellite (i.e., GPM, SMAP, and TROPOMI) and in situ observations collected at/round Harvard Forest and the CRN-Millbrook site during the SMAPVEX22 campaign were analyzed along with WRF-Chem results during a precipitating event associated with a frontal passage that occurred from late 13 July to early 14 July 2022. This event caused sharp increases in SSM around 14 July in Massachusetts (by >0.06 $m^3$ $m^{-3}$) and parts of the eastern New York (by ~0.02 $m^3$ $m^{-3}$), as well as drastic changes in air temperature (by up to ~5 K decreases at the surface) and other meteorological fields. These changes in SSM and meteorological conditions contributed to abrupt $O_3$ reductions of up to 30 ppbv. Baseline simulation without DA failed to reproduce the strong daily SSM variability at site-to-regional scales (Fig. 13b). After enabling the SMAP DA, Noah-MP SSM in Massachusetts and the eastern New York increased remarkably on 14 July (Fig. 13c), better matching the observed quantities. Along the southern New York-Connecticut as well as the northern New York-Vermont borders, the slightly drier conditions on 14 July are also better represented in Noah-MP with the implementation of SMAP DA (Fig. 13a–c). The enhancements in soil wetness resulted in altered precipitation characteristics, a bit cooler surface soil/air, thinner

atmospheric boundary layer, suppressed biogenic VOC and soil $NO_y$ emissions as well as $O_3$ formation while deposition accelerated. Lightning emissions were also sensitive to the DA-induced SM changes. Consequently, above many Connecticut River watershed areas, WRF-Chem $NO_2$ columns dropped (Fig. 13e–f). Due to increased upwind pollution contributions whereas weakened local emissions and production, both enhancements and reductions by up to ~4 ppbv in daytime surface $O_3$ levels (not shown in figures) are found in the New England region (40.5–43.1°N, 70.0–74.0°W). Across the New England region, WRF-Chem daytime surface $O_3$ performance for 14 July was improved in 31 out of 50 of the model grids where AQS data were available, with the largest improvement of nearly 2 ppbv. It is also highlighted that the various processes SM can impact help shape the vertical profiles of $NO_2$ and other chemical species, a critical ancillary data for calculating the air mass factor that is needed to convert slant columns to vertical columns in satellite retrievals (Lorente et al., 2017) and derive averaging kernels (AKs, Eskes and Boersma, 2003). At Harvard Forest, the vertical distributions of $NO_2$ as well as their responses to SMAP DA changed rapidly during this event (Fig. 13g–h), despite the minor change in $NO_2$ column. It is suggested that cautions are taken when attributing the mismatches between TROPOMI and models (with AKs, that indicate lower TROPOMI sensitivity towards the surface) over the scenes where $NO_x$ near the surface and aloft may both be significant. Also, productions, interpretations, and applications of satellite $NO_2$ retrievals could benefit from evaluating and tuning their model-based *a priori* profiles with in situ measurements of $NO_2$ vertical distributions under various environments.

Figure 14a–c illustrates that, on a larger timescale, SMAP DA effectively narrowed the Noah-MP wet biases in July 2022-July 2018 SSM differences in Canada's Ontario (croplands) as well as the dry biases in Virginia (forests) that may have resulted from inaccurate representations of meteorological drought conditions. WRF-Chem weather fields, biogenic VOC, soil $NO_y$ and lightning emissions, and deposition processes all responded to the DA-induced changes in the model's land ICs. The July 2022-July 2018 differences of WRF-Chem $NO_2$ columns and surface $O_3$ over these regions became closer to (by as high as ~50% and >4 ppbv, respectively) what TROPOMI and AQS observations indicate (Fig. 14d–i). Notably, the SMAP DA flipped the sign of surface $O_3$ interannual differences over the northern Virginia, for which region the DA had strong impacts on the modeled surface $O_3$ in both July 2018 and July 2022 (Fig. S18). The remaining modeled-observed $NO_2$ and $O_3$ discrepancies over some of the northern states and coastal North Carolina, which are highly correlated because of the dominating $NO_x$-limited regime, can also be explained by uncertainties in the model's chemical BCs and wind fields.

These analyses demonstrate that microwave satellite SM DA can improve the modeled SM dynamics at daily-to-interannual timescales. Similar findings were previously reported by Draper and Reichle (2015) where SM from the X-band (sensitive to top ~1 cm soil) Advanced Microwave Scanning Radiometer-Earth Observing System was assimilated at only four sites, but not on regional scales for forested regions where SM retrievals have been considered challenging and need validation. It is also shown in this work that the DA adjustments to LSMs' SM fields can positively impact weather and chemistry fields from their coupled atmospheric models, benefiting our interpretations and prediction skills of air pollutants' distributions and

temporal changes which can in turn help advance satellite retrievals. It is important to note that SSM-atmosphere coupling strengths vary strongly in space and time, influenced by the evolution of local hydrological regimes. As 2022-2018 SSM and surface air temperature differences show strong negative correlations of -0.78 (Fig. S19), the land DA impacts on WRF-Chem's atmospheric chemistry fields were in some part through adjusting the weather, as indicated in Figs. 14j–l and S18. For the times/locations that SSM and atmosphere coupling strengths are weak, land DA is anticipated to impact the modeled atmospheric chemistry fields mostly via the direct control of land surface on natural emissions and deposition.

### 3.3.2 Irrigation approaches

Based on the three sets of simulations representing full-, reduced-, and no-irrigation scenarios (Section 2.3.2), the impacts of irrigation on surface $O_3$ concentrations, CUO and $O_3$ injury to vegetation, as well as Nr deposition were quantified (Fig. 15). Across the domain, $O_3$ perturbs gross primary productivity more strongly (up to 20–30%) than transpiration (mostly <10%), and therefore reduces the vegetation water use efficiency. Its reductions to leaf biomass over the stressed irrigated lands in the Carolinas in late June 2022 are estimated to be <5% under all three scenarios. Under the limited- and no-irrigation conditions, $O_3$-induced crop yield losses were reduced over irrigated areas by up to ~2%, a result of lowered SM (Fig. S20) and deposition fluxes despite the enhanced soil/air temperatures, soil $NO_y$ emissions and surface $O_3$ concentrations (by up to ~10 ppbv). This result supports and extends the findings from previous coarse-resolution modeling (Mills et al., 2018) and observational (Harmens et al., 2019) studies. The period-integrated $O_3$ stomatal uptake increased slightly outside of the irrigated land due to higher $O_3$ being transported away from the irrigated areas. Over/near the irrigated areas, the estimated total Nr deposition would also be lower under reduced- and no-irrigation scenarios by more than 50%, which would be below possible CL thresholds, as less irrigation would result in stronger atmospheric mixing and reduced SM although soil $NO_y$ emissions would increase. These impacts on Nr deposition over most of the irrigated lands are important also according to Student's $t$-tests comparing the base and sensitivity simulations ($p<0.05$). The impacts of irrigation on Nr deposition over non-irrigated areas are rather noisy and more intense than on $O_3$, where Student's $t$-tests comparing Nr deposition from different simulations gave larger-than-0.05 $p$ values. These sensitivities away from irrigated lands still highlight the complex net effects of irrigation-induced changes in land surface and meteorological conditions on a group of species with substantially different properties undergoing various atmospheric processes.

Compared with long-term offline LSM simulations forced by independently produced $O_3$ data, evaluations of $O_3$ vegetation impacts using coupled modeling systems like WRF-Chem with land surface feedback to regional weather and atmospheric chemistry being accounted for are more realistic. Nevertheless, such approaches are hundreds of times more computationally expensive and may be subject to uncertainty from the atmospheric model. Survey- and satellite-based irrigation types and water use information, including wastewater use that may impact plant growth, nutrient supply and soil environments (Aman et al., 2018), direct stationary and/or airborne measurements of water, carbon, energy, air pollutants' concentrations and fluxes, as well as plant traits within and outside of irrigated areas for variable hydroclimatic conditions, would help evaluate

and improve irrigation modeling and the model-based holistic assessments of irrigation impacts on regional environments that could assist with forming pollution mitigation and ecosystem adaptation strategies for future.

### 3.3.3 Transboundary pollution

Periodically, distant pollution sources make strong environmental impacts on the Northeast and Mid-Atlantic US states. For
example, during the 2023 SARP-East campaign, JPSS-1/CrIS observed high $O_3$ and low CO on 13 June; and high $O_3$, CO, and PAN on the following days of the same week (Fig. 16) when elevated $NH_3$ columns and aerosol optical depths were also observed from space by multiple instruments (not shown). These data suggest that long-range transported stratospheric air and Canadian wildfire plumes reached the eastern US.

As indicated by the stratospheric $O_3$ tracer of the chemical BC model WACCM, the 13 June stratospheric intrusion event associated with a frontal passage enhanced lower tropospheric $O_3$ by as high as 30–40 ppbv along the northeast corridor, which helps explain the spike at ~700 hPa (>30 ppbv $O_3$ enhancement) in the SARP-East RRC ozonesonde profile for that day (Fig. 17a–b). The WACCM-estimated stratospheric impact on surface $O_3$ in our WRF-Chem domain is only ¼–1/3 of its impact on free tropospheric $O_3$ (Fig. S21), consistent with prior knowledge that stratospheric impacts on the US East are
often confined at higher altitudes while surface $O_3$ remains low (Ott et al., 2016). Thick Canadian wildfire plumes that moved into the study region, dramatically enhanced $O_3$ spanning a wide altitude range (i.e., from >900 hPa to ~600 hPa) above the RRC site on several days of that week (Fig. 17b). Under the strong influences of Canadian fires, $O_3$ in the US outflows during that week was close to that in the air sampled ~two decades ago along the East Coast (Cooper et al., 2005). Ozonesondes also indicate that air quality improved remarkably in the following week, with $O_3$ from the surface to ~700 hPa
nearly 40 ppbv lower (Fig. 17b).

The WRF-Chem baseline and "Sen" sensitivity simulations were evaluated with $O_3$ observations during 13–16 June 2023. Overall, the baseline and "Sen" simulations moderately well reproduced the daytime surface $O_3$ patterns and diurnal cycles observed at AQS sites during the events, with RMSEs of ~7 ppbv (Figs. 17c–k and S21). The negative mean biases of 1–2
590 ppbv in the modeled daytime peak $O_3$ (Fig. 17c) can be explained by the model's incapability of accurately representing the stratospheric $O_3$ influences. The choice of WACCM's fire emission input had minor impacts on WRF-Chem daytime surface $O_3$ averaged across the domain throughout the episode but enhanced/reduced WRF-Chem's daily daytime surface $O_3$ by up to ~10 ppbv on grid scale (Figs. S21–S22).

The extremely high transported background aerosols and their precursors due to Canadian wildfires, along with upwind US pollution, interacted with meteorological and land surface fields (e.g., radiation, temperature, clouds, precipitation, and surface wetness) that are relevant to evapotranspiration and photosynthesis (see also discussions on Asian anth pollution impacts in Huang et al., 2020, and references therein), dry deposition velocity and wet deposition coefficient, and secondary

pollutant formation. The baseline and "Clean BC" cases together indicate that, although under highly polluted conditions, dry deposition velocities are overall reduced (Fig. S23) and photochemistry activities are weakened, the period-integrated CUO and mean total Nr deposition fluxes are enhanced as the excessive amount of imported pollution significantly elevated surface $O_3$ and Nr concentrations (Fig. 18). During this period, daily $O_3$ stomatal uptake and the mean total Nr deposition overland are ~2% and ~26% higher than their 2023 seasonal-mean values, respectively. Comparisons of baseline and "Sen" simulations results show that the modeled grid-scale $O_3$ stomatal uptake and total Nr deposition are sensitive to the choice of WACCM's fire emission input (Fig. S22). This set of sensitivity analysis not only supports the findings about fire emission impacts on deposition from offline air quality modeling studies (e.g., Koplitz et al., 2021), but also stresses the importance of accounting for aerosol radiative effects in assessing ecosystem impacts of pollutants from biomass burning and other sources, which will be investigated further on multiple spatiotemporal scales in an Hemispheric Transport of Air Pollution phase 3 multimodel experiment (Whaley et al., 2024, https://doi.org/10.5194/gmd-2024-126, in review).

Previous work has focused strongly on the impacts of long-range transport of pollution from Asia and the stratosphere, as well as regional pollution transport, on the western US $O_3$ trends (e.g., Cooper et al., 2012; Huang et al., 2013; Lin et al., 2017; Miyazaki et al., 2022). This case study demonstrates that extra-regional pollution can also compromise the efforts of improving air quality via controlling local and regional emissions over the eastern US. Possibly linked to climate change, such highly polluted events occurred more frequently during the 2023 warm season. For example, driven by hot and dry conditions, the Canadian 2023 wildfire season had the largest area burned in history (https://cwfis.cfs.nrcan.gc.ca/ha/nfdb, last access: 23 July). Due to Canadian wildfire impacts, there were at least two other known extreme air pollution events over the eastern US in June 2023 and more in the other months of the season (Fig. S24). Such events exerted controls on surface-atmosphere exchange processes and perturbing the long-term changes in $O_3$, Nr and other chemical compounds. More accurate and consistently-configured chemical BC models or reanalysis products, preferably at higher resolutions with a more complete list of prognostic and diagnostic variables, are essential for further regional-scale modeling investigations on such events and their contributions to trends/variability. Addition of stratospheric tracers and accurate, time-varying upper chemical boundary conditions to regional models, assisted with $O_3$ profile measurements from commercial aircraft, sondes, and Lidar networks, are expected to be also helpful for diagnosing and/or reducing errors in the simulations of some of such events.

## 4 Summary and suggested future directions

Based on WRF-Chem model simulations and multiplatform observations, this paper discussed Nr and $O_3$ concentrations and fluxes during 2018–2023 in the northeastern and mid-Atlantic US, most of which fell into $NO_x$-limited and transitional chemical regimes. Effective local emission controls resulted in evident decreases in $NO_2$ and surface $O_3$ concentrations, with the reduced human activities during the COVID lockdowns also contributing to their low values and the overall stronger surface $O_3$-$NO_2$ column correlations in 2020. Current polar-orbiting satellites take snapshots of $NO_2$ columns only at a

particular time of day, such as in the early afternoon when surface $NO_2$ experienced their daily lows. With this sparse temporal sampling, TROPOMI did not miss the general $NO_2$ interannual and seasonal variability and filled in the extremely large horizontal gaps between surface AQS observations most of which are in/near urban regions and positively biased. The WRF-Chem simulation described here has been extended into 2024, running on a routine basis, to support refined analyses concerning the subdaily variability of $NO_2$ and other variables along with data from geostationary satellite missions such as the TEMPO.

The declines in $NO_y$ emissions and concentrations were roughly consistent with the temporal changes in $NO_y$ deposition, which were balanced out by the overall rising agricultural and total $NH_x$ emissions and deposition. The changes in $NO_y$ and $NH_x$ deposition together shaped the interannual variability in Nr deposition in contrast to the clearer downward trends in $O_3$ vegetation uptake that reduced plants' water use efficiency and caused biomass/crop yield losses by a few percent. Certain hotspots of Nr deposition in North Carolina may have continued to exceed the CL thresholds for herbaceous plants and trees in literature, while the productivity of northern forests may have remained to be nitrogen-limited. Referring to the conservative lichen CL thresholds in literature, widespread lichen CL exceedances likely occurred persistently. Integrating nitrogen dynamics into LSMs could help improve their performance on land surface states as well as carbon, water, and energy fluxes, and further, the representations of Nr and $O_3$ deposition processes and their interactions in coupled modeling systems. Standard versions of Noah-MP, including what was used in this work, represent nitrogen stress by applying constant foliage nitrogen factors (<1) in maximum carboxylation rate calculations (Niu et al., 2011). Following the JULES and Community Land Model, Cai et al. (2016) started to add nitrogen dynamics to Noah-MP. Running offline, their updated model yielded more accurate net primary productivity and evapotranspiration, and that may also be embedded into Earth system models in future, with the magnitudes and spatiotemporal variability of its Nr inputs (e.g., from deposition and fertilizer applications) being improved with the aid of atmospheric chemistry model routines or/and observations. Other areas for improvements include assimilating additional Earth observations (e.g., rootzone SM and terrestrial water storage); developing and applying high-quality, spatially and temporally varying $CO_2$ forcings for Noah-MP; and tuning parameters that represent $O_3$ vegetation impacts for various types of plants at different growth stages.

With updated model parameterizations and anth emissions, the used WRF-Chem system performed stably and remarkably better on eastern US surface $O_3$ than many of those in literature. This paper highlighted that, temporal variability of Nr and $O_3$ concentrations and fluxes on subregional-to-local scales were partially driven by hydrological variability that can be influenced by precipitation and controllable human activities such as irrigation. Like deposition processes, biogenic soil Nr and VOC emissions that are highly sensitive to various climatic factors and plants' physiological conditions, as well as extra-regional sources (e.g., dense wildfire plumes from the western US and Canada, and $O_3$-rich stratospheric air), have been playing increasingly important roles in controlling pollutants' budgets in this area as local emissions went down. These outcomes based on this particular WRF-Chem system have implications for updating other modeling systems.

It is worth noting that, urban emissions and air pollutants can be transported to and deposited into rural and remote regions, which may better be modeled at finer resolutions (e.g., urban scale at 1–4 km or street-to-building scales) with the urban landscapes and human influences on urban vegetation and soil properties being more carefully handled. Finer model

resolutions may also allow more processes, such as convection, to be explicitly resolved, potentially leading to more accurate precipitation and deposition results. Successful finer-resolution simulations would require accurate inputs and observational constraints at similar resolutions. To better inform the designs of mitigation and adaptation strategies, it is highly recommended to continue evaluating and improving the parameterizations and inputs relevant to various sources and processes in seamlessly coupled multiscale Earth system models using laboratory and field experiments in combination with

satellite DA. Further improved Earth system model results are expected to in turn benefit remote sensing communities, for example, via serving as the retrieval *a priori* profiles for different types of environments.

**Code and data availability**

NASA-Unified Weather Research and Forecasting model (https://nuwrf.gsfc.nasa.gov/software, last access: 6 February 2024) output of $O_3$ and other key variables will be shared via Zenodo with the final version of this manuscript. Remote

sensing and in situ data sets can be downloaded from: https://doi.org/10.5067/4DQ54OUIJ9DL (O'Neill et al., 2021), https://doi.org/10.5067/GPM/IMERGDF/DAY/07 (Huffman et al., 2023); https://doi.org/10.5067/MHH8R0UZ5BMJ (Bowman, 2022a); https://doi.org/10.5067/JL1HT3NGEAW3 (Bowman, 2022b); https://doi.org/10.5067/6HTQB4F81S08 (Bowman, 2022c); https://www-air.larc.nasa.gov/cgi-bin/ArcView/listos (Janz, 2020); https://doi.org/10.5065/4F4P-E398 (NCEP, 2004); and https://aqs.epa.gov/aqsweb/airdata/download_files.html (US EPA, 2024, last access: 1 October 2024).

Gridded TROPOMI data have been submitted by Isabelle De Smedt to TOAR-II Tropospheric Ozone Precursors Focus Group repository.

**Author contributions**

Overall study design and paper writing: MH leading, all participated in reviewing and editing

Design, execution, or discussions of model simulations: MH, GRC, SVK, ABG

Satellite data production, validation, delivery, and analysis: KWB, IDS, AC, MHC, MH

Field campaign deployments and data analysis: AC, MHC, SJJ, RMS, AMT, NMF, RJS, JDB, ATJ, MH

**Competing interests**

The authors declare that they have no competing interests.

**Acknowledgements**

NASA SMAP, SMAPVEX22 and LDAS sponsored part of the work. A contribution was made to this work at the Jet Propulsion Laboratory, California Institute of Technology, under a contract with NASA. We acknowledge the excellent leadership of TOAR-II Tropospheric Ozone Precursors and Deposition Focus Group leads. We thank Antonin Soulie and team for helping with the CAMS global anth emissions, and Kyle DeLong for participating in collecting SM data in Harvard Forest during SMAPVEX22 that are used in this study. Color palettes in Crameri et al. (2020) are used in this paper.

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

**Figures**

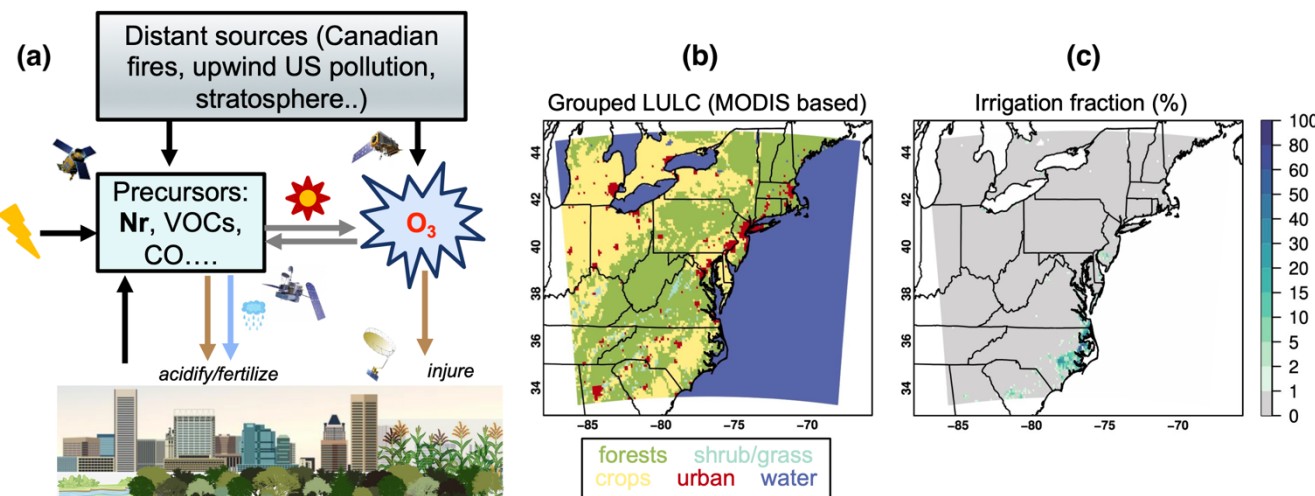

**Figure 1: (a) A simplified schematic representation of Nr-O$_3$ relationships in the Earth systems; (b) model domain and the grid-dominant land use/land cover (LULC) classifications, grouped from the original 20-category International Geosphere-Biosphere Programme-modified Moderate Resolution Imaging Spectroradiometer (MODIS) using the same criteria as in Huang et al. (2022); and (c) irrigation fraction information required in the irrigation scheme. The grouped LULC is used for reporting potential O$_3$ and Nr ecosystem impacts in Section 3.2, and approximately 32%, 24%, 1%, 3%, and 40% of model grids belong to the grouped**
**forests, crops, shrub/grass, urban, and water category, respectively.**

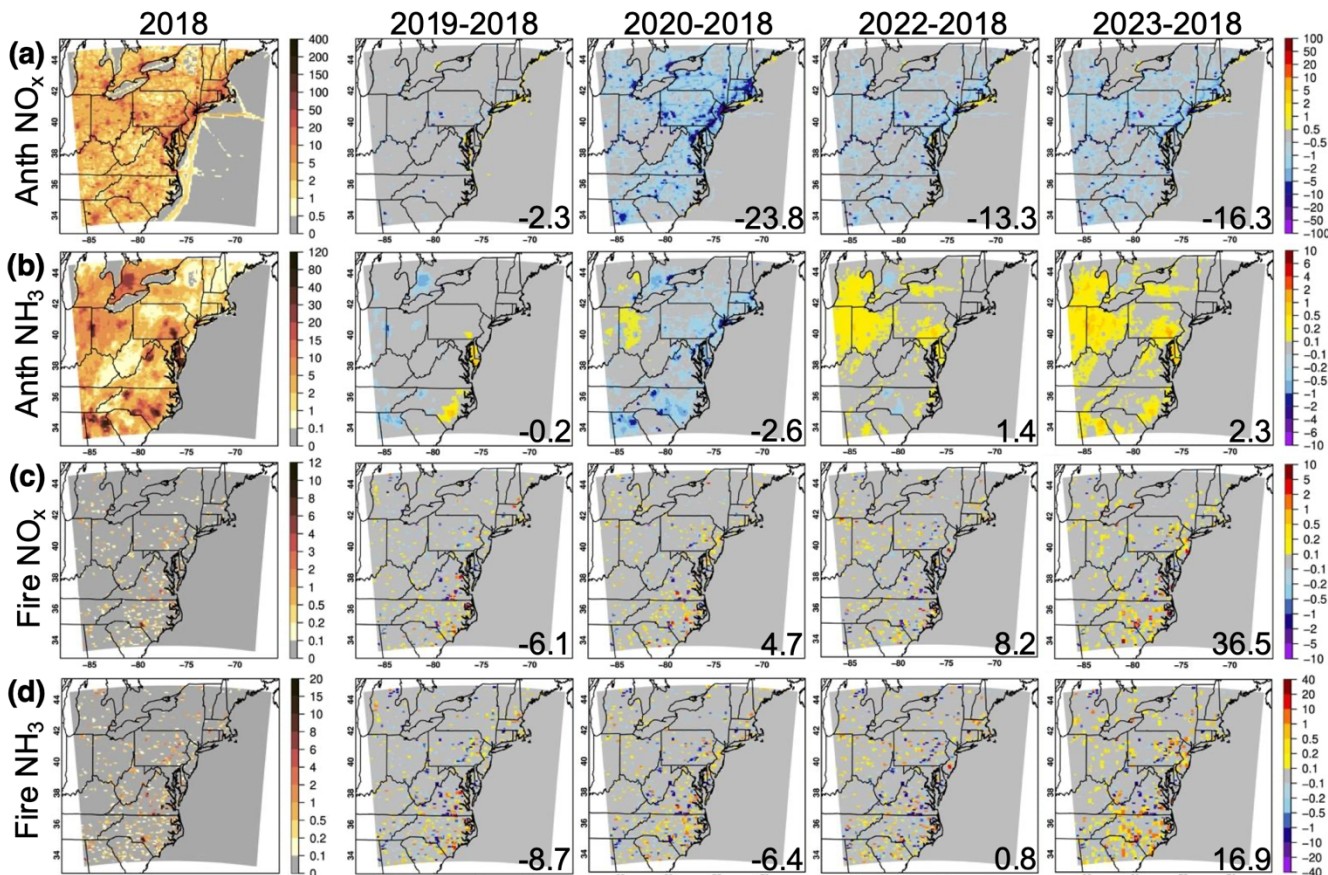

**Figure 2:** (a;b) Anthropogenic (Anth) and (c;d) biomass burning (fire) (a;c) $NO_x$ and (b;d) $NH_3$ emissions for MJJ 2018 and the differences between MJJ of each of the following years and 2018, in mol km$^{-2}$ h$^{-1}$. Numbers at the corners of the difference plots indicate the % changes relative to MJJ 2018.

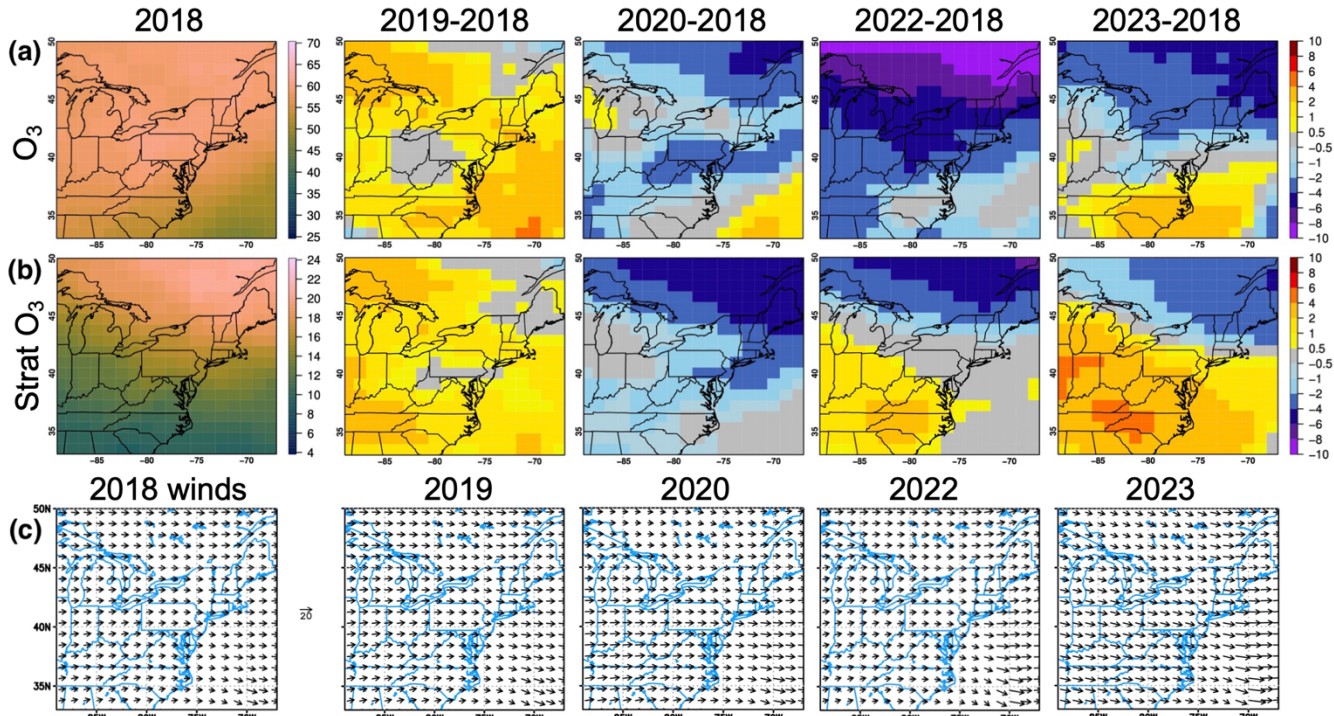

**Figure 3: MJJ ~600–800 hPa (a) total and (b) stratospheric O₃ and their interannual differences in ppbv, and (c) wind fields for each year's MJJ, from WRF-Chem's chemical boundary condition models. Stratospheric O₃ impacts on the surface are indicated in Fig. S2.**


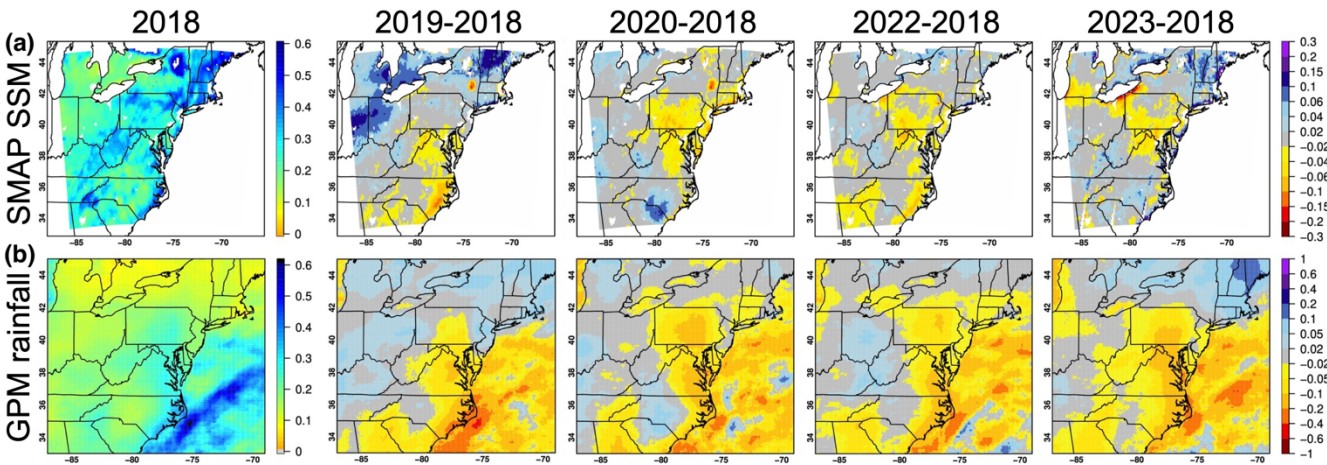

**Figure 4: (a) SMAP morning-time SSM (m³ m⁻³) on WRF-Chem grids and (b) GPM/IMERG precipitation (mm h⁻¹) on its native grid for MJJ 2018 (left) and the differences between MJJ of each of the following years and 2018. SMAP measures the globe every 2–3 days and GPM daily global-coverage products are used for this work. SMAP data are not available during 20 June–22 July 2019 due to instrument outages; and the ESA CCI version 8.1 SM product indicates qualitatively similar MJJ 2019-2018 variability.**


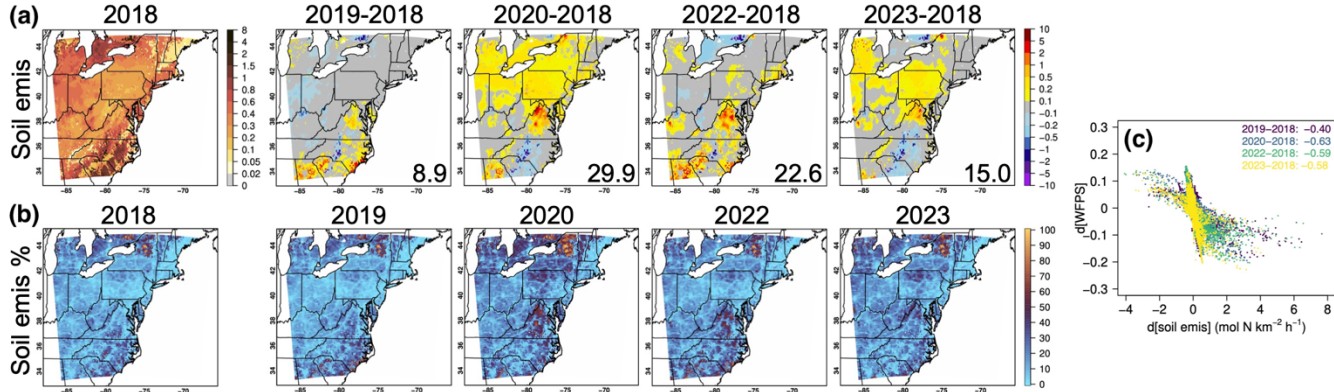

**Figure 5: (a)** Modeled soil NO and HONO emissions (mol N km$^{-2}$ h$^{-1}$) and **(b)** soil NO and HONO emission % contributions to total anth+fire+soil NO$_y$ emissions. Model results are averaged for MJJ 2018, shown together with the differences between MJJ of each of the following years and 2018. Numbers at the corners of the soil emission difference plots in (a) indicate the % changes relative to MJJ 2018. The scatterplot in (c) indicates relationships between the interannual differences in water-filled pore space (WFPS, whose spatial patterns are shown in Fig. S5) and soil NO$_y$ emissions including their correlation coefficients in the upper-right legend ($p \ll 0.01$).

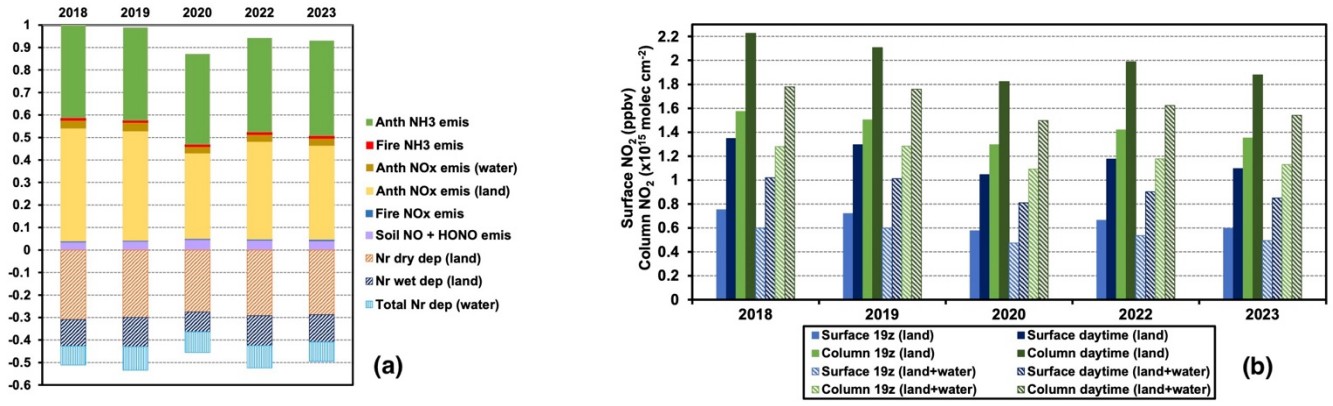

**Figure 6: (a)** Emission and deposition fluxes by year, scaled to MJJ 2018 total emissions; **(b)** Domain-wide MJJ-average surface and column NO$_2$, summarized for early-afternoon (19 UTC) and daytime, and for land and all model grids. Water and land model grids are defined in Fig. 1b.

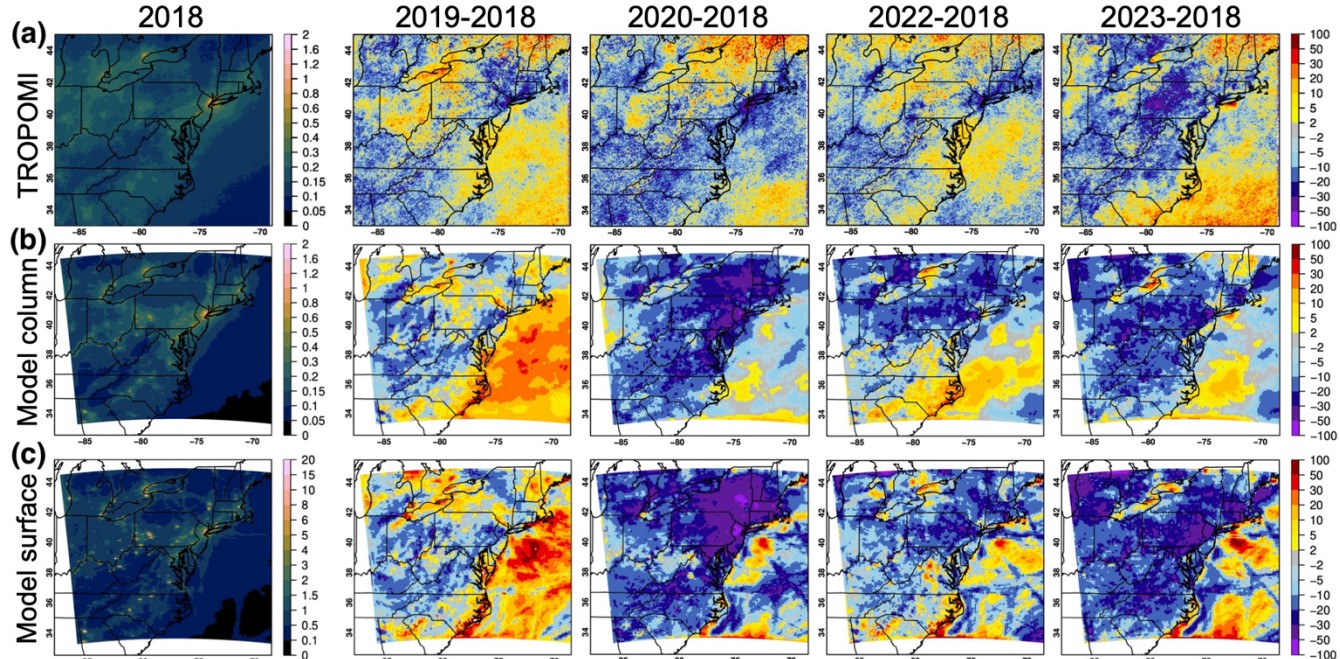

**Figure 7: (a) TROPOMI and (b) WRF-Chem NO₂ columns; and (c) WRF-Chem surface NO₂ at 19 UTC. Results are averaged for MJJ 2018 (left, in ×10¹⁶ molec. cm⁻² for column NO₂ and ppbv for surface NO₂) and shown together with the % differences between MJJ of each of the following years and 2018.**

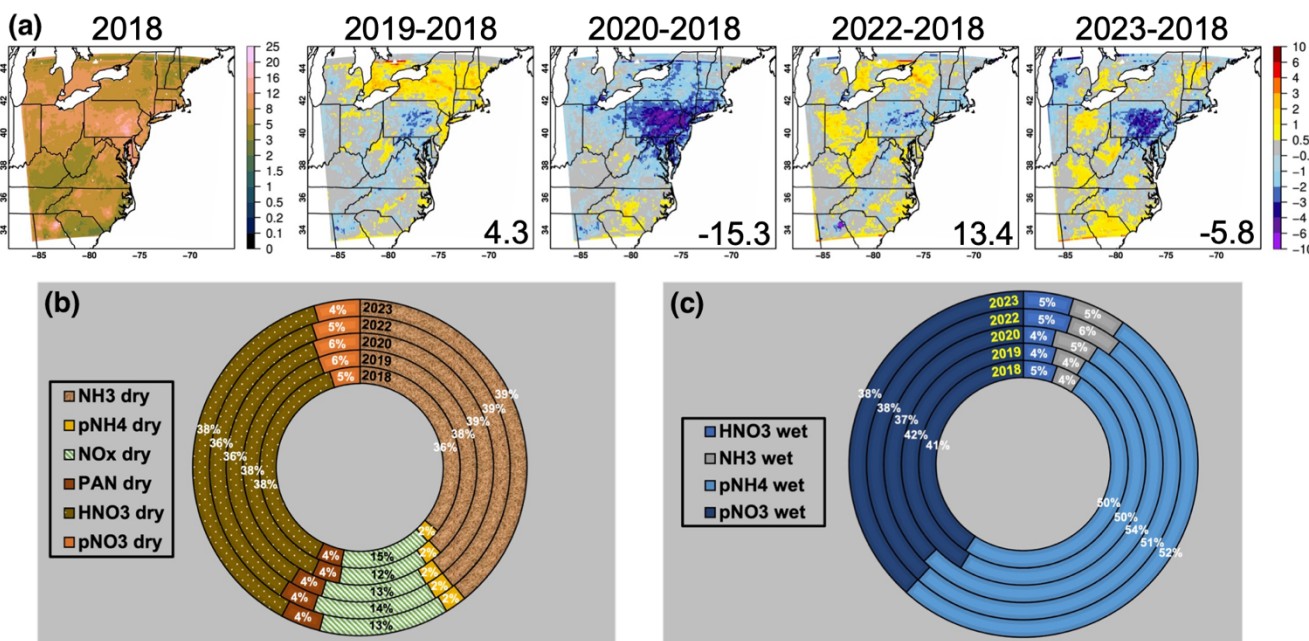

**Figure 8: (a) Modeled MJJ 2018 total Nr deposition overland and differences between MJJ of each of the following years and 2018 in kgN ha⁻¹ a⁻¹; and speciation of modeled (b) dry and (c) wet deposition fluxes by year, where prefix "p" indicates particle.**

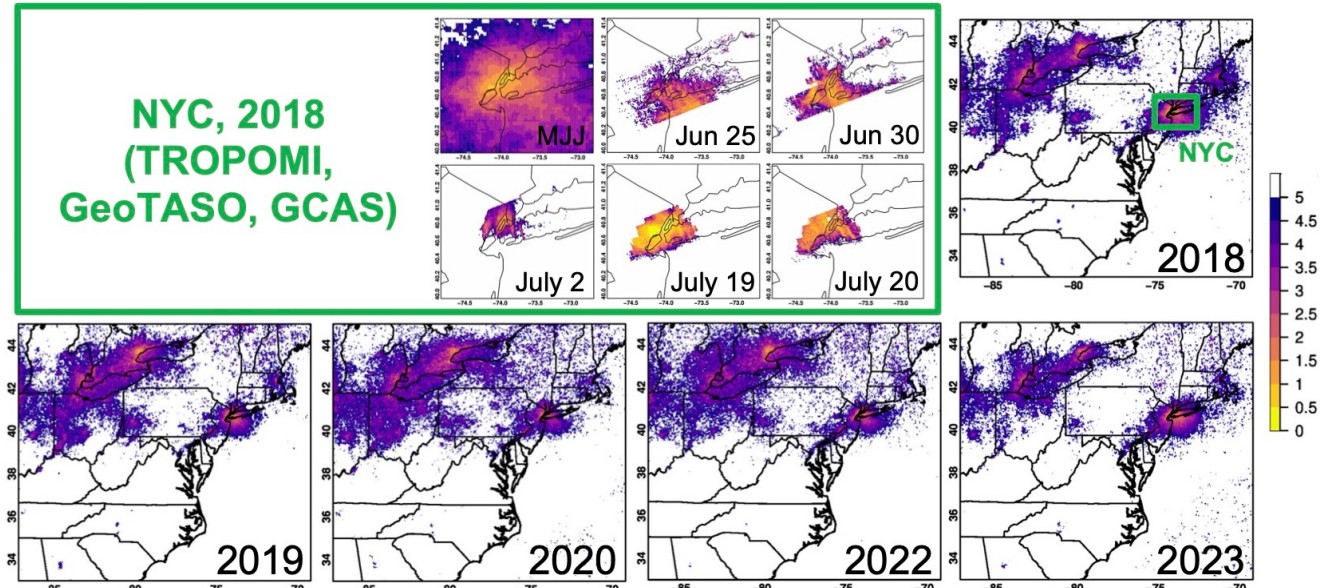

**Figure 9:** TROPOMI (MJJ 2018–2023), GeoTASO (25 and 30 June 2018) and GCAS (2, 19 and 20 July 2018) HCHO/NO₂ ratios. GeoTASO and GCAS both took measurements over the Greater New York City (NYC) several times during the sampling days which indicate subdaily variability in HCHO, NO₂ and their ratio. Their measurements closest to 19 UTC are used here.

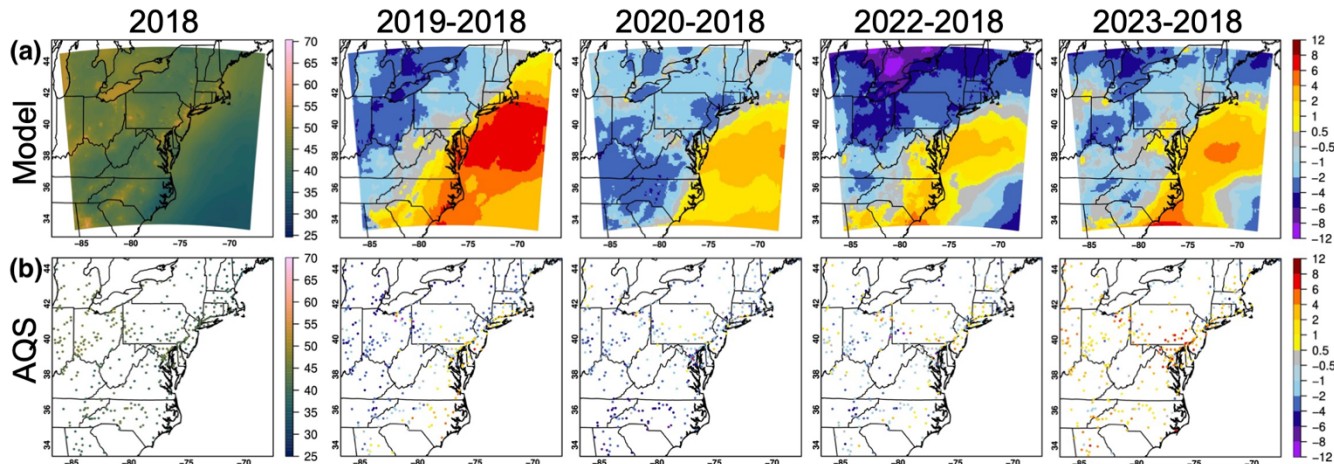

**Figure 10:** (a) WRF-Chem modeled and (b) AQS daytime surface O₃. Results are averaged for MJJ 2018, shown together with the differences between MJJ of each of the following years and 2018, all in ppbv. Observations from the AQS sites having <10% missing data for each year were used for evaluation. Model vs. AQS RMSEs (number of grids having collocated observations) for 2018, 2019, 2020, 2022, 2023 are 5.6 (375), 6.5 (377), 5.9 (373), 4.8 (370), and 4.0 (381), respectively.

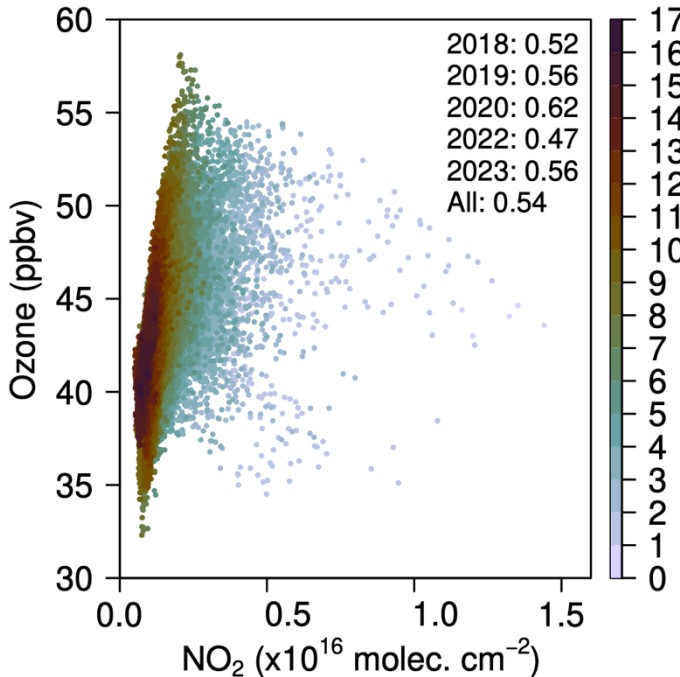


**Figure 11: Scatterplot indicating the relationships between WRF-Chem modeled daytime surface O₃ and 19 UTC NO₂ column during MJJ 2018–2023 for all terrestrial model grids, colored by column HCHO/NO₂ ratios. Their correlation coefficients ($p \ll 0.01$) are indicated in the corner legend by year.**

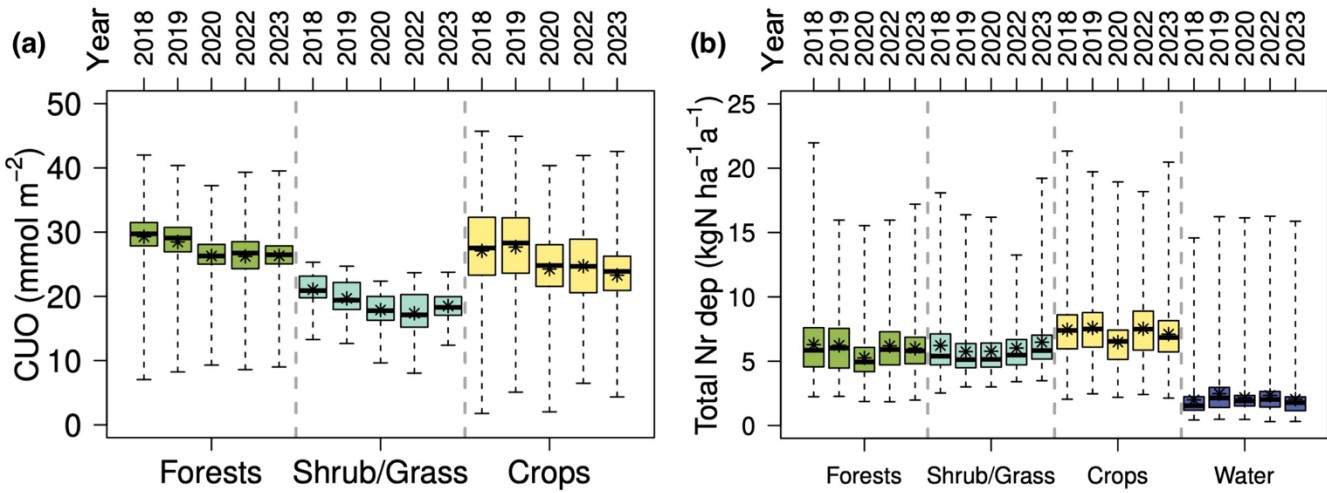


**Figure 12: Box-and-Whisker plots of (a) CUO and (b) mean total Nr deposition fluxes for MJJ 2018–2023 by the grouped surface types defined in Fig. 1b.**

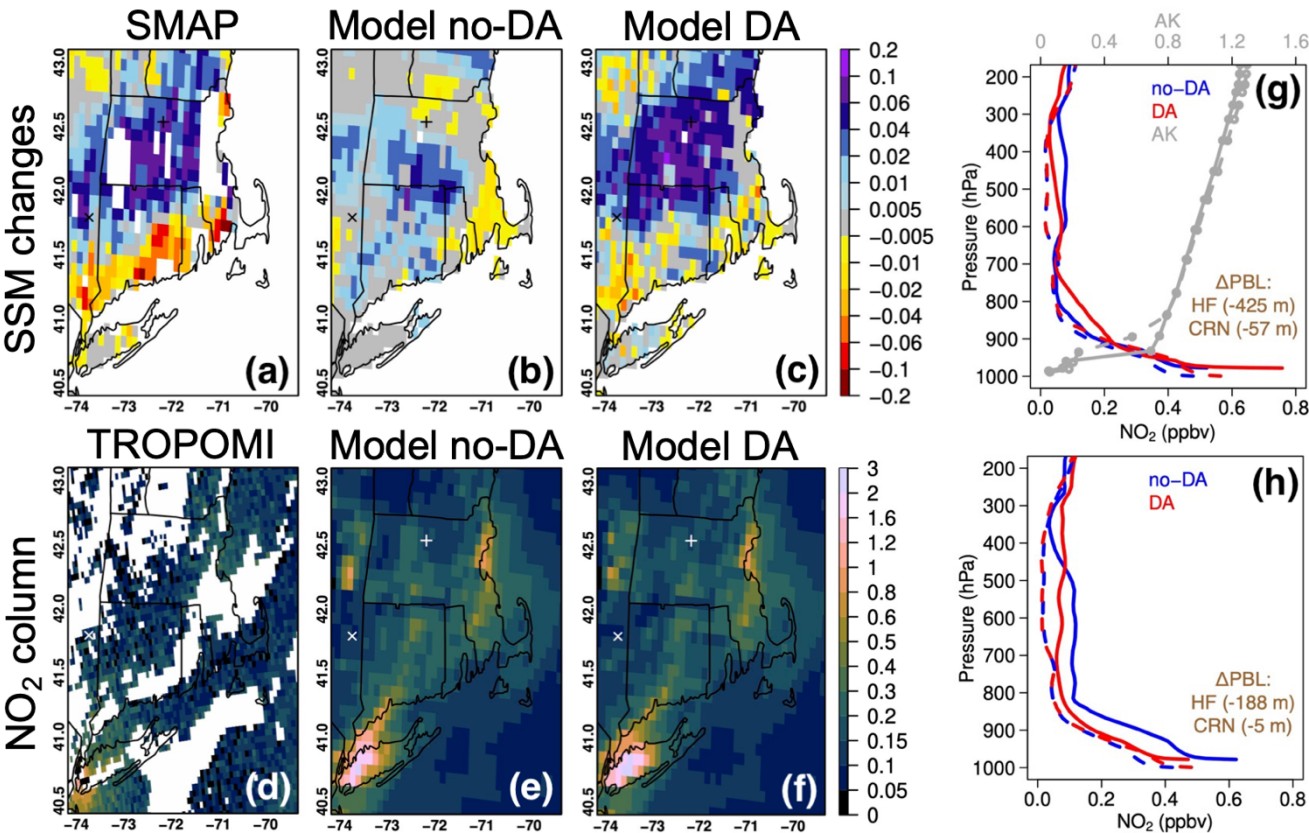

**Figure 13:** (a–c) 14 July-11 July SSM (m³ m⁻³) changes indicated by bias-corrected SMAP, free-running and SMAP-constrained Noah-MP results; (d–f) 14 July TROPOMI NO₂ columns (×10¹⁶ molec. cm⁻²) collected between 18–19 UTC, free-running and SMAP-constrained WRF-Chem results at 18 UTC; NO₂ vertical profiles from free-running and SMAP-constrained WRF-Chem at Harvard Forest (HF, solid line) and CRN-Millbrook (dash line) at (g) 18 UTC and (h) 19 UTC on 14 July, along with the impact of SMAP DA on modeled boundary layer height (PBL) as well as TROPOMI averaging kernels (AK) on TROPOMI's *a priori* model grid. The white + and × signs in (a–f) denote the locations of HF and CRN-Millbrook where in situ precipitation and SSM data are also analyzed. Ground-based SSM measurements on 11 July and 14 July near SMAP overpasses are 0.170±0.059 and 0.245±0.080 m³ m⁻³ at HF, and 0.067 and 0.086 at CRN-Millbrook, respectively. Precipitation and ground-based O₃ fields on 11 and 14 July are shown in Fig. S17.

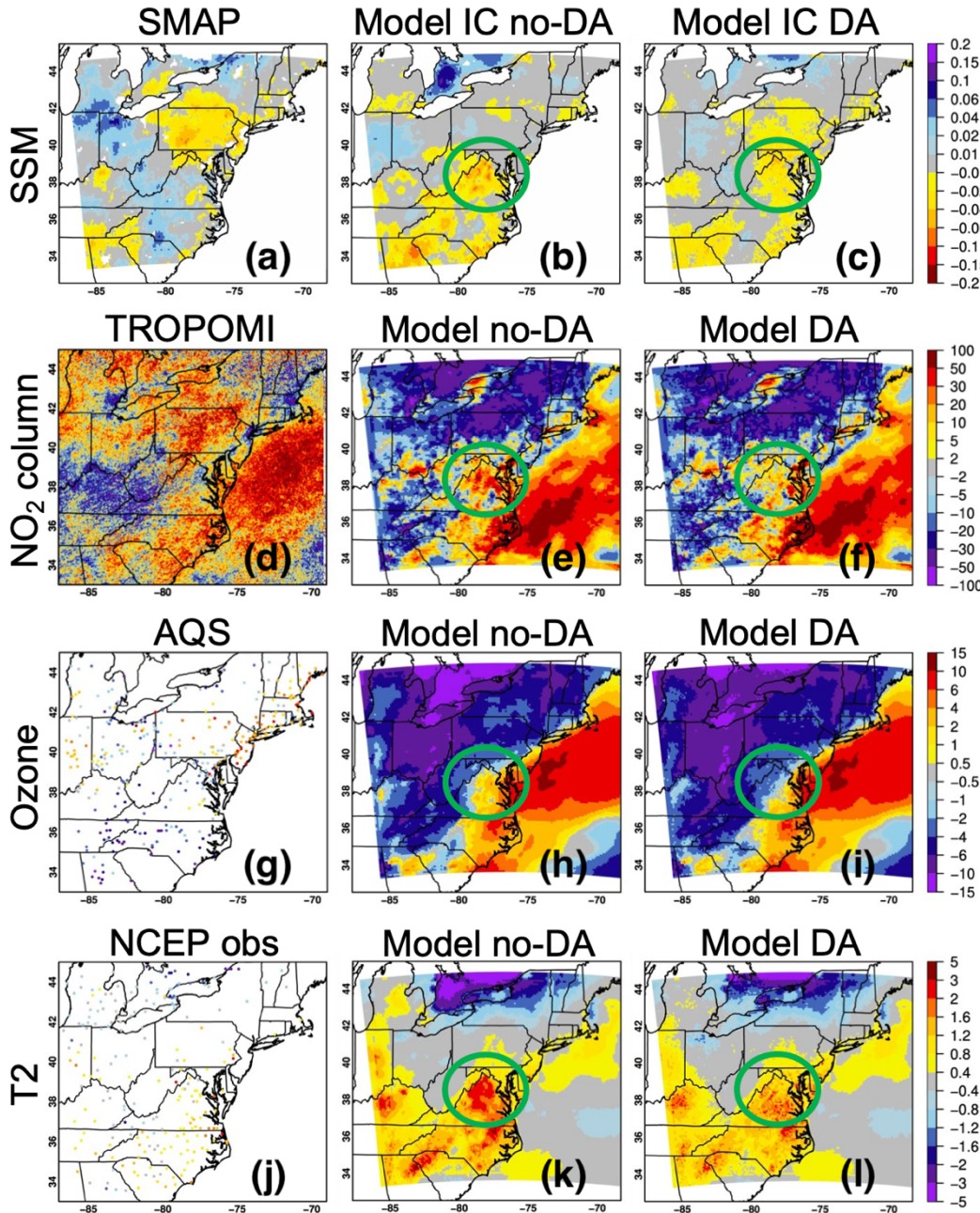

**Figure 14: July 2022-July 2018 monthly differences in (a–c) SSM (m³ m⁻³) indicated by bias-corrected SMAP, free-running and SMAP-constrained WRF-Chem initial conditions (ICs); (d–f) early afternoon NO₂ columns (%) based on TROPOMI, free-running and SMAP-constrained WRF-Chem results; (g–i) daytime surface O₃ concentrations (ppbv) based on AQS observations, free-running and SMAP-constrained WRF-Chem results; and (j–l) daytime surface air temperature (K) based on the National Centers for Environmental Prediction (NCEP) Surface Observational Weather Data product, free-running and SMAP-constrained WRF-Chem results. Green circles highlight areas in/around Virginia where improvements in WRF-Chem land ICs notably improved the weather, NO₂ and O₃ fields. Additional information on the SMAP data assimilation (DA) impacts is included in Fig. S18.**


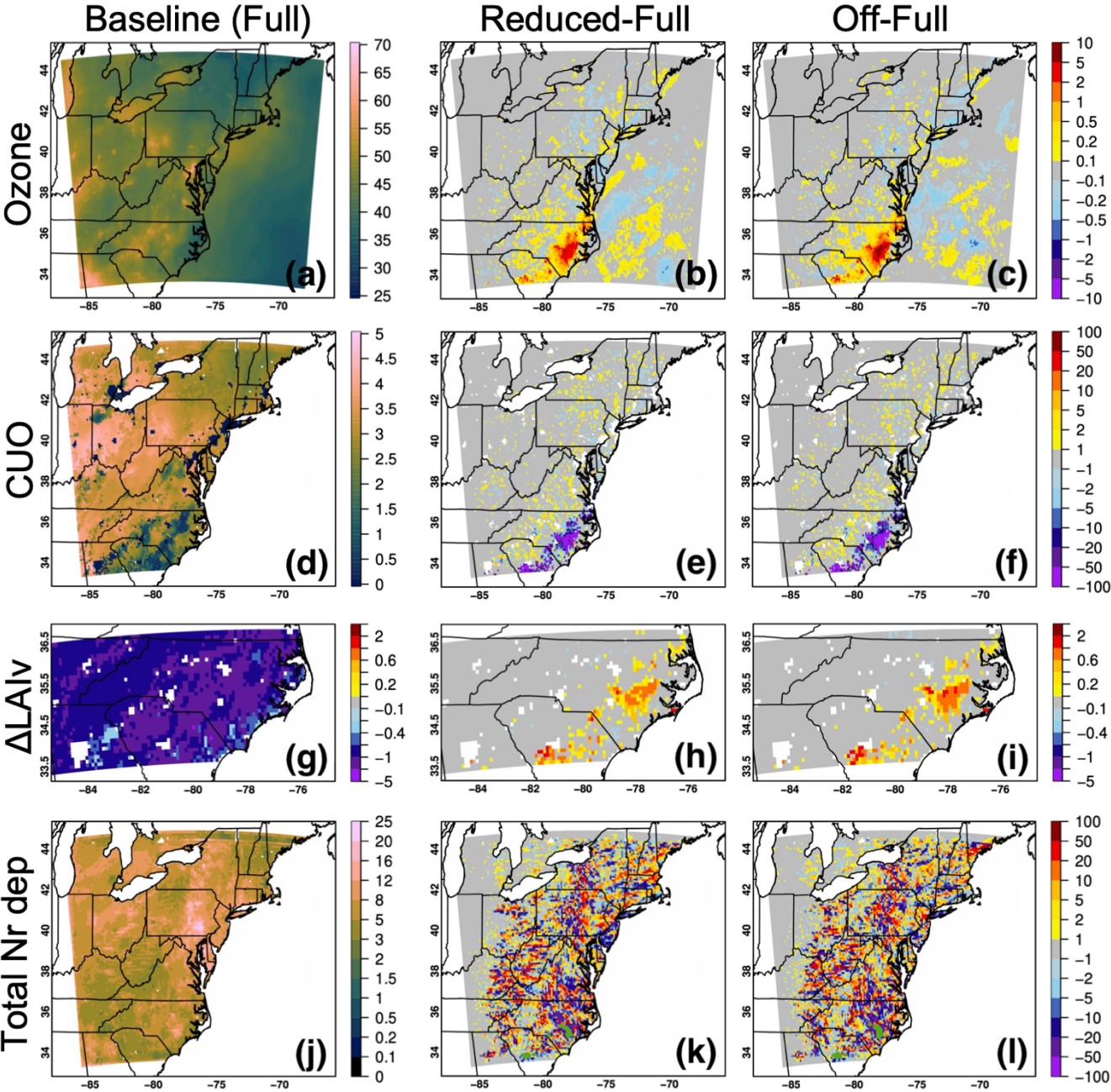

**Figure 15:** (a) Daytime surface $O_3$ concentration (ppbv, with the RMSE relative to AQS data of ~5.6 ppbv); (d) period-cumulated $O_3$ stomatal uptake (mmol m$^{-2}$); (g) $O_3$ impacts on leaf biomass (%) over irrigated areas in/around the Carolinas; and (h) total Nr deposition overland (kgN ha$^{-1}$ a$^{-1}$) from the baseline simulation during 21–30 June 2022, and (b;c;e;f;h;i;k;l) their sensitivities to adjustments in irrigation schemes. Sensitivity results are in ppbv for surface $O_3$ concentration, and in % for all other plots. Green areas in (k;l) marked the grids where Student's $t$-tests comparing Nr deposition from the baseline and sensitivity simulations gave smaller-than-0.05 $p$ values.

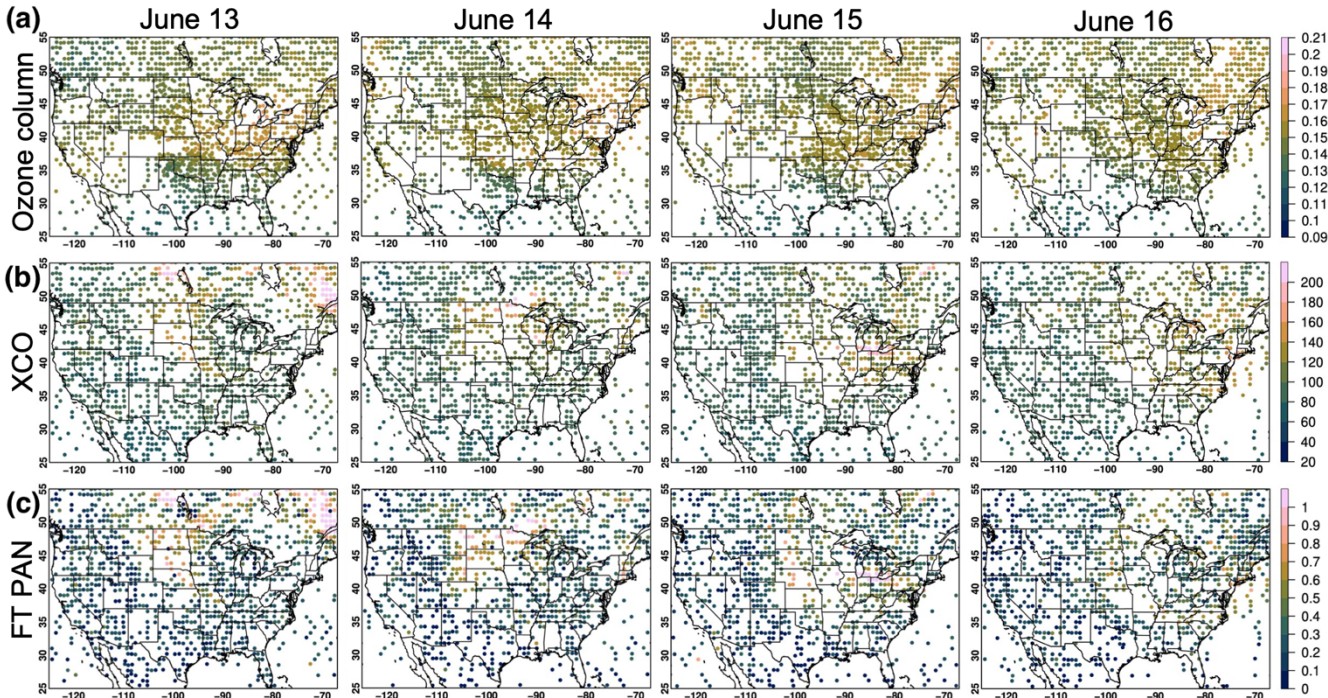

**Figure 16:** JPSS-1/CrIS observed (a) $O_3$ columns (mol m$^{-2}$); (b) column-averaged CO mixing ratios (ppbv); and (c) column-averaged PAN mixing ratios (ppbv) for the free troposphere between 825 and 215 hPa, during 13–16 June 2023.


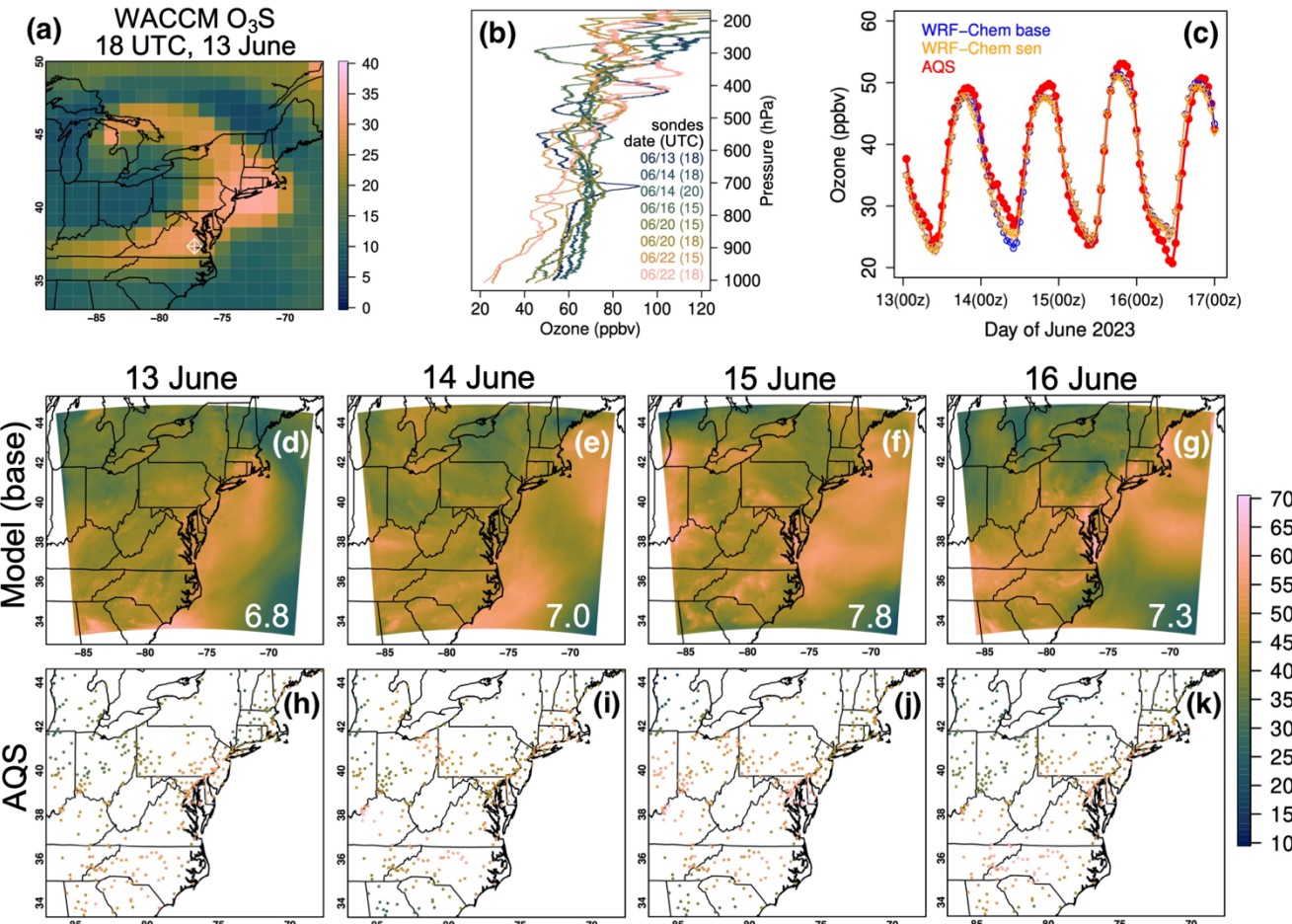

**Figure 17: (a)** WACCM model stratospheric $O_3$ tracer (ppbv) results at ~700h Pa at 18 UTC of 13 June 2023, with location of the RRC site being indicated by a white diamond; **(b)** Ozonesonde profiles launched from the RRC; **(c)** Timeseries of the domain-mean observed and WRF-Chem modeled hourly surface $O_3$ during 13–16 June 2023 at AQS sites; and daytime surface $O_3$ concentrations (ppbv) on 13–16 June 2023 from **(d–g)** the WRF-Chem baseline simulation and **(h–k)** AQS sites. WRF-Chem vs. AQS RMSEs (ppbv) are indicated in the lower-right corners of **(d–g)**.


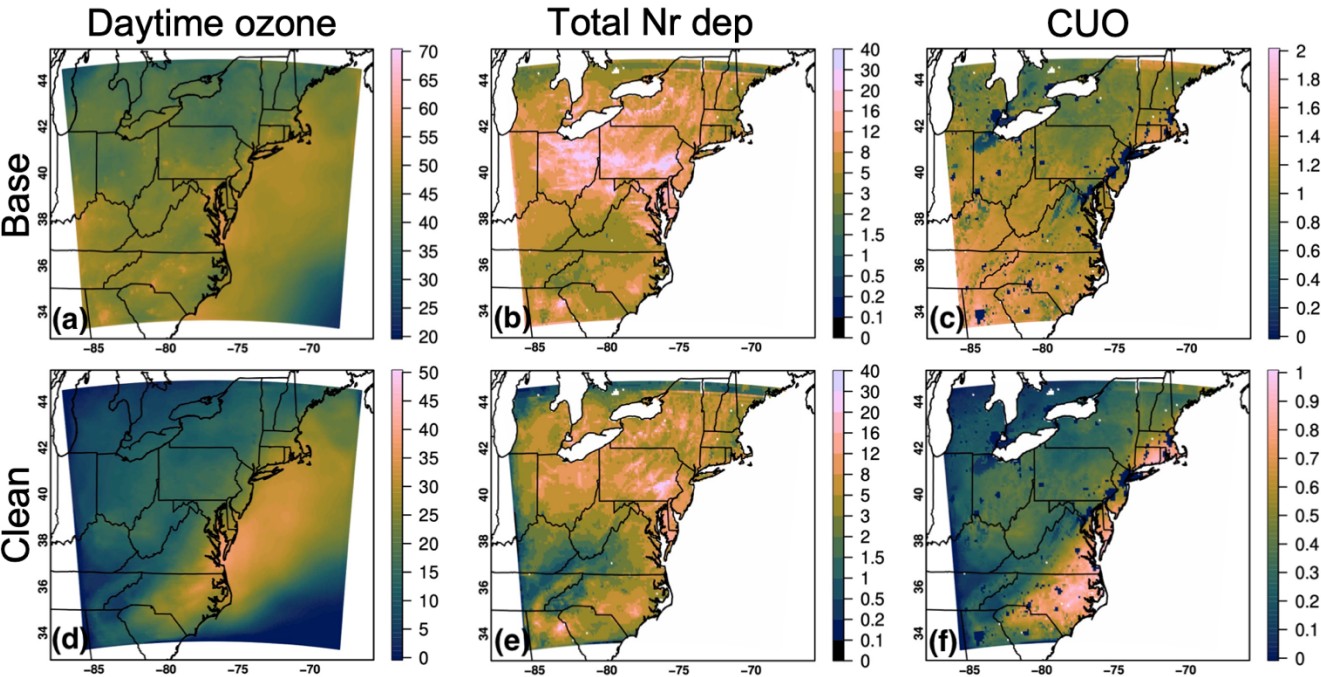

Figure 18: (a;d) Daytime surface $O_3$ concentrations (ppbv); (b;e) total Nr deposition overland (kgN ha$^{-1}$ a$^{-1}$); and (c;f) period-cumulated $O_3$ stomatal uptake (mmol m$^{-2}$) during 13–16 June 2023 from the (a–c) baseline simulation and (d–f) sensitivity simulation with clean chemical BCs.