# Peer review of "Reactive nitrogen in and around the northeastern and Mid-Atlantic US: sources, sinks, and connections with ozone"

_EGUsphere, 2024_

## Author Comment (AC1)

Comments by Owen R. Cooper (TOAR Scientific Coordinator of the Community Special Issue) on:

**Reactive nitrogen in and around the northeastern and Mid-Atlantic US: sources, sinks, and connections with ozone**

Authors: Huang, M., Carmichael, G. R., Crawford, J. H., Bowman, K. W., De Smedt, I., Colliander, A., Cosh, M. H., Kumar, S. V., Guenther, A. B., Janz, S. J., Stauffer, R. M., Thompson, A. M., Fedkin, N. M., Swap, R. J., Bolten, J. D., and Joseph, A. T.

EGUsphere [preprint], https://doi.org/10.5194/egusphere-2024-484, 2024
Discussion started: 22 Feb 2024; Discussion closes May 1, 2024

This review is by Owen Cooper, TOAR Scientific Coordinator of the TOAR-II Community Special Issue. I, or a member of the TOAR-II Steering Committee, will post comments on all papers submitted to the TOAR-II Community Special Issue, which is an inter-journal special issue accommodating submissions to six Copernicus journals: ACP (lead journal), AMT, GMD, ESSD, ASCMO and BG. The primary purpose of these reviews is to identify any discrepancies across the TOAR-II submissions, and to allow the author teams time to address the discrepancies. Additional comments may be included with the reviews. While O. Cooper and members of the TOAR Steering Committee may post open comments on papers submitted to the TOAR-II Community Special Issue, they are not involved with the decision to accept or reject a paper for publication, which is entirely handled by the journal's editorial team.

**General Comments:**

TOAR-II has produced two guidance documents to help authors develop their manuscripts so that results can be consistently compared across the wide range of studies that will be written for the TOAR-II Community Special Issue. Both guidance documents can be found on the TOAR-II webpage: https://igacproject.org/activities/TOAR/TOAR-II

*The TOAR-II Community Special Issue Guidelines*: In the spirit of collaboration and to allow TOAR-II findings to be directly comparable across publications, the TOAR-II Steering Committee has issued this set of guidelines regarding style, units, plotting scales, regional and tropospheric column comparisons, tropopause definitions and best statistical practices.

*Guidance note on best statistical for TOAR analyses*: The aim of this guidance note is to provide recommendations on best statistical practices and to ensure consistent communication of statistical analysis and associated uncertainty across TOAR publications. The scope includes approaches for reporting trends, a discussion of strengths and weaknesses of commonly used techniques, and calibrated language for the communication of uncertainty. Table 3 of the TOAR-II statistical guidelines provides calibrated language for describing trends and uncertainty, similar to the approach of IPCC, which allows trends to be discussed without having to use the problematic expression, "statistically significant".

**Specific Comments:**

My comments are minor, as follows:

1) When discussing the impact of COVID on ozone it would be helpful to cite a new paper published in the TOAR-II Community Special Issue. Putero et al. (2023) show that ozone decreased in 2020 at high elevation sites across the western USA, and also at four high elevation sites in the eastern USA. Another

paper that is relevant to the COVID period is Steinbrecht et al. (2021) who show that ozone also decreased in the free troposphere of northern mid-latitudes.

2) When discussing long-term ozone trends across the USA, the papers that are currently available in the peer-reviewed literature are out-of-date, as they all seem to end in 2014 or 2015. However, EPA provides regular ozone trend updates on the following webpage:
https://www.epa.gov/air-trends/ozone-trends
They focus on the 98[th] percentile, or the annual 4[th] highest MDA8 ozone value.

3) As summarized in the TOAR-II "Guidance note on best statistical for TOAR analyses", the TOAR community is abandoning the expression "statistically significant" for the reasons described by Wasserstein et al. (2019).  Please follow these recommendations and replace "statistically significant" on line 396 by describing your confidence in this result.

**References:**

Putero, D., Cristofanelli, P., Chang, K.-L., Dufour, G., Beachley, G., Couret, C., Effertz, P., Jaffe, D. A., Kubistin, D., Lynch, J., Petropavlovskikh, I., Puchalski, M., Sharac, T., Sive, B. C., Steinbacher, M., Torres, C., and Cooper, O. R.: Fingerprints of the COVID-19 economic downturn and recovery on ozone anomalies at high-elevation sites in North America and western Europe, Atmos. Chem. Phys., 23, 15693–15709, https://doi.org/10.5194/acp-23-15693-2023, 2023.

Steinbrecht, Wolfgang, Dagmar Kubistin, Christian Plass-Dülmer, Jonathan Davies, David W. Tarasick, Peter von der Gathen, Holger Deckelmann, Nis Jepsen, Rigel Kivi, Norrie Lyall, Matthias Palm, Justus Notholt, Bogumil Kois, Peter Oelsner, Marc Allaart, Ankie Piters, Michael Gill, Roeland Van Malderen, Andy W. Delcloo, Ralf Sussmann, Emmanuel Mahieu, Christian Servais, Gonzague Romanens, Rene Stübi, Gerard Ancellet, Sophie Godin-Beekmann, Shoma Yamanouchi, Kimberly Strong, Bryan Johnson, Patrick Cullis, Irina Petropavlovskikh, James W. Hannigan, Jose-Luis Hernandez, Ana Diaz Rodriguez, Tatsumi Nakano, Fernando Chouza, Thierry Leblanc, Carlos Torres, Omaira Garcia, Amelie N. Röhling, Matthias Schneider, Thomas Blumenstock, Matt Tully, Clare Paton-Walsh, Nicholas Jones, Richard Querel, Susan Strahan, Ryan M. Stauffer, Anne M. Thompson, Antje Inness, Richard Engelen, Kai-Lan Chang, Owen R. Cooper (2021), COVID-19 Crisis Reduces Free Tropospheric Ozone Across the Northern Hemisphere, Geophysical Research Letters, 48, e2020GL091987. https://doi.org/10.1029/2020GL091987

Wasserstein, R.L., Schirm, A.L. and Lazar, N.A., 2019. Moving to a world beyond "p< 0.05". The American Statistician, 73(sup1), pp.1-19
https://www.tandfonline.com/doi/pdf/10.1080/00031305.2019.1583913

---

## Author Comment (AC2)

**Please see below our point-by-point response (in blue) to reviewers' comments (in black). Quoted text from the revised manuscript is *in italic*.**

**Response to RC1**

**General Comments**

Overall, the paper is interesting and uses an updated version of WRF-Chem with the Noah-MP LSM (i.e., the NASA-Unified Weather Research and Forecasting model with online chemistry). The paper covers relatively important aspects of the complex interactions between changes in reactive nitrogen (Nr) fluxes and ozone formation in the eastern U.S., and the relationships and implications of such changes with changing land/biosphere-atmosphere interactions. The paper highlights such interactions that are relatively less appreciated in the community in regards to land surface data/processes, Nr fluxes, and ozone. Indeed, the paper attempts to cover a lot of complex land-atmosphere-chemistry topics. However, many arguments and discussions, appear only cursory with minimal to no supporting scientific evidence/analyses or quantitative assessments provided for justification. I provide explicit comments below that pertain to some of these issues in the paper, and suggestions to improve them. Also, the manuscript also appears flawed in its grammar and sentence structure, where a thorough proof-read of the English writing should have been done prior to submission. To return the review comments in a timely manner, I can only provide the detailed grammatical and sentence structure errors in the Abstract and Introduction sections, with general issues in other sections that need to be thoroughly revised (see Technical Corrections below). The grammatical errors are persistent through the remaining manuscript, and it is imperative that the manuscript is thoroughly checked for grammar and English writing in all sections before submission of the revised version. Otherwise, in my opinion, the paper is not acceptable for publication.

Thanks for the overall positive comment on the paper content. Multiple authors who are native English speakers edited previous versions of this manuscript. More information supporting our arguments has been added. Careful proofreading of the revised manuscript has been conducted.

**Specific Comments**

**2 Methods**

Line 103: Model spinup description of "late April" is not specific enough. What was the exact number of days used for spinup in April?
"*Late April*" has been changed to "*25 April*".

Lines 144-145. OK, but seems that omission of detailed fertilizer inputs and bi-directional ammonia fluxes (BIDI-NH3) in the modeling system is an oversight considering all other advances. The BIDI-NH3 approach has been shown to have significant impacts on overall deposition of Nr species and aerosols. Can this in any way be rectified, or at least discussed in terms of model uncertainty using only a unidirectional approach in areas of significant ammonia fluxes (e.g., agricultural lands)? This becomes an issue later in the results section of the paper (i.e., Lines 351-353).

The decision of not applying a BIDI-$NH_3$ approach in this case was carefully made. We are aware of the potential uncertainty introduced by BIDI approaches which can result from the assumptions in calculating stomatal and ground compensation points, and good information for adequately addressing that uncertainty is lacking, especially at grid scale. The cited review papers summarize numerous published methods and datasets related to emission potential (for calculating compensation points) and compensation points – the data are overall highly variable and extremely limited for the US regions, making it difficult for them to be broadly applied to the US ecosystems we study here. Based on a few previous studies (previous L351-353), the uncertainty of not applying a BIDI approach was estimated to be a few percent over source regions. This estimated uncertainty can be highly uncertain because emission potential, compensation points, and $NH_3$ fluxes in these studies were not evaluated.

The WRF-Chem modeled $NH_3$ fields have been evaluated with data from the NADP/Ammonia Monitoring Network (AMoN, see revised Table S2 and Fig. S13) which to a certain degree can indicate the suitability of the applied $NH_3$ deposition method in this system.

The impacts of fertilizer inputs were not omitted but included in the CAMS anth emission input that is introduced in the following paragraph.

Lines 161-163:  Agreed, but there has been recent work that addresses and quantifies this issue of using satellite retrievals to infer NOx emissions. See (Silvern et al., 2019; https://acp.copernicus.org/articles/19/8863/2019/) and Zhen et al., 2021; https://pubmed.ncbi.nlm.nih.gov/34149109/). Indeed there is much uncertainty given the increasingly large role of background NO2 emission sources.
Elguindi et al. (2020) discussed results in Silvern et al. (2019) to reach the point being referred to here. To make this clear, we added "*references therein*". Both Silvern et al. (2019) and Qu et al. (2021) are based on a global model which may not capture $NO_x$ lifetime and budgets so well as fine-resolution systems. A few main sources of uncertainty were discussed by the authors of those papers.

Line 164:  Please describe more about what plume rise approach is used and uncertainties. Plume rise has a major impact on wildfire emission transport.
The default WRF-Chem plume rise scheme based on Freitas et al. (2007) was used, as introduced by Grell et al. (2011). Grell et al. (2011) is now cited. The impact of plume rise vs. horizontal winds, emission intensity, and chemistry on wildfire emission transport and climate feedback on event-to-multiyear time scales is an active research topic being addressed by various communities (e.g., Veira et al., 2015).

Freitas, S. R., Longo, K. M., Chatfield, R., Latham, D., Silva Dias, M. A. F., Andreae, M. O., Prins, E., Santos, J. C., Gielow, R., and Carvalho Jr., J. A.: Including the sub-grid scale plume rise of vegetation fires in low resolution atmospheric transport models, Atmos. Chem. Phys., 7, 3385–3398, https://doi.org/10.5194/acp-7-3385-2007, 2007.

Grell, G., Freitas, S. R., Stuefer, M., and Fast, J.: Inclusion of biomass burning in WRF-Chem: impact of wildfires on weather forecasts, Atmos. Chem. Phys., 11, 5289–5303, https://doi.org/10.5194/acp-11-5289-2011, 2011.

Veira, A., Kloster, S., Schutgens, N. A. J., and Kaiser, J. W.: Fire emission heights in the climate system – Part 2: Impact on transport, black carbon concentrations and radiation, Atmos. Chem. Phys., 15, 7173–7193, https://doi.org/10.5194/acp-15-7173-2015, 2015.

Lines 171-172:  And due to rising agricultural, plant fertilizer application and emissions.
Agreed. This information is included in the description here.

Line 191:  I think this section is better titled: "Chemical observations from satellites, aircraft, and ozonesondes."
Changed as suggested.

Lines 215 – 231: While this is a very useful assimilation and comparison using satellite SM products, I think some uncertainty should be explained regarding lack of deeper soil moisture observations and understanding, and implications for drought.  The top-level soil measurements at first 5 cm, woefully neglects the important impacts of rootzone SM on drought.
The impact of assimilating satellite surface SM on SM in deeper soil layers in part depends on the surface–subsurface coupling strengths of the used land systems (Kumar et al., 2009; Huang et al., 2022).

Land data assimilation that integrates surface SM (e.g., L-Band SMAP) as well as rootzone SM (e.g., P-Band AirMOSS and SNOOPI; thermal infrared ALEXI) and terrestrial water storage (e.g., GRACE and GRACE-FO) will likely lead to even more robust results. This is however not always true – see Figs. 8 and 9 in Hain et al. (2012). Related suggestions have been added to Section 4.

Hain, C. R., Crow, W. T., Anderson, M. C., and Mecikalski, J. R.: An ensemble Kalman filter dual assimilation of thermal infrared and microwave satellite observations of soil moisture into the Noah land surface model, Water Resour. Res., 48, W11517, https://doi.org/10.1029/2011WR011268, 2012.

Lines 219-223: This is a very broad statement, and no definitely understanding on how results presented here either qualitatively or quantitatively agree/disagree with the NADM (e.g., spatiotemporal comparisons).  Please revise and be more explicit.
The NADM, as well as the USDM, is developed based on many sources of information by rotating authors. Therefore, it is partially subjective and designed to indicate various types of droughts - meteorological, agricultural, and hydrological. Inevitably, comparisons between SM data and the NADM/USDM have been typically qualitative.

Here, we now include state-level NADM drought extents and their temporal variability (Table S1) and discuss them together with Fig. 4a and the added standard deviations of SMAP SM in Fig. S3. For the period of case study #2, the Vegetation Drought Response Index maps are now shown together with a USDM map in Fig. S4.

Lines 243-245: Understood that long-term direct chemical flux measurements are limited, but what about the CASTNET (dry deposition) and NADP (wet deposition) networks?

As discussed in Huang et al. (2022) and references therein, dry deposition fluxes from the CASTNET dataset are partially model-based, which have known limitations and biases against eddy covariance flux measurements as well as fluxes estimated using other methods. Taking Referee #2's suggestion, we instead evaluated the modeled PM speciation with data from CASTNET and AQS sites, the modeled $HNO_3$ concentrations with CASTNET data, and the modeled $NH_3$ concentrations with NADP/AMoN observations. In addition, we compared the diurnal cycles of the modeled $O_3$ dry deposition velocity $v_d$ at Harvard Forest for the study period with flux measurements reported in literature for previous decades. In the Supplement (Fig. S7), we mentioned: "*At Harvard Forest, WRF-Chem MJJ and measured $v_{d,o3}$ (during 1990–2000 June-July-August-September, Clifton et al., 2017) display similar diurnal cycles, with their daytime maxima and nighttime minima of 0.8–1.0 and <0.3 cm s$^{-1}$, respectively*".

The NADP/National Trends Network (NTN) nitrogen and sulfur wet deposition fluxes, as well as precipitation, have been analyzed and included in the revised Supplement.

Clifton, O. E., Fiore, A. M., Munger, J. W., Malyshev, S., Horowitz, L. W., Shevliakova, E., Paulot, F., Murray, L. T., and Griffin, K. L.: Interannual variability in ozone removal by a temperate deciduous forest, Geophys. Res. Lett., 44, 542–552, https://doi.org/10.1002/2016GL070923, 2017.

**3 Results**

Lines 318-339:  I find parts of this section very speculative, and qualitative, where it is difficult to follow the emphasis (also due to writing issues, noted below).
This paragraph mostly describes the spatiotemporal variability of surface and column $NO_2$ based on quantitative results presented in several figures (Figs. 6, 7, and previous S5-S8/current S7-S10). The descriptions on the passive lightning tracer (no chemical reactions involved) and surface $NO_2$ measurements (known to be positively biased) have to be qualitative. We cannot find anything more specific from this reviewer's later comment on these lines.

Lines 343-346:  So are the differences in dry deposition contributions a result of overestimated wet deposition in other literature/models, or underestimations of wet deposition based on WRF-Chem? This is a confusing and rather contradictory argument.
Both are reasons. Nr wet deposition was underestimated in WRF-Chem (see added evaluation in Table S2 and Fig. S11) and overestimated in some other studies/models, so contributions of dry deposition to the total in WRF-Chem are larger than in those other studies/models and possibly overestimated. We adjusted these two sentences and inserted descriptions on the new wet deposition evaluation results here.

Lines 341-365:  Again, this paragraph largely compares the results from this modeling system based on WRF-Chem to other literatures, and has some rather speculative arguments.  I think much of this paragraph could be trimmed, improved writing (see below), and improved discussion.  Its plausible this section is more an assessement of the WRF-Chem modeling evaluated generally, and mainly qualitatively against other models, and some measurements.  Not sure what is new here.

It's very common and important to compare results from one's own study with literature. We have added and discussed the new evaluation results and broken down this paragraph.

Lines 367-379:  I think this could be improved by directly comparing the modeled spatial Nr deposition to critical load thresholds for different vegetation types and how it has changed from 2018-2023.

Simkin et al. (2016) critical load thresholds, as well as the lower and higher limits of the 95% confidence interval of these thresholds, are compared with the modeled Nr deposition fluxes for a subset of model grids. The estimated exceedances are now presented in Fig. S16.

Lines 393-394:  I also think the much higher (lower) correlation coefficients against NO2 (HCHO) in year 2020 deserve to be highlighted and discussed briefly.

Fig. 11 shows that in 2020 $O_3$-$NO_2$ correlations were higher whereas $O_3$-HCHO correlations were much lower than those in other years. This model-based result suggests an overall stronger $NO_x$-sensitive regime in 2020 partly due to COVID impacts. This sentence has been changed to: "*Daytime surface $O_3$ concentrations exhibit more robust spatial correlations with early afternoon (19 UTC) $NO_2$ columns than HCHO columns, especially for 2020 due to COVID, with correlation coefficient r of 0.54 (0.62) and 0.40 (0.07) for all years (year 2020), respectively (Fig. 11)*". Related sentences in the abstract and Section 4 have also been updated.

Lines 425-428:  Here it would be good to briefly describe the controlling parameters on decreasing CUO in the past 2018-2023, and that projected to continue to decrease in the future climate.  Seems much uncertainty here, and justification is needed for discussion.  If the eastern U.S. is projected to become wetter climate in the future, it would suggest increasing CUO, unless ozone concentrations decrease enough in proportion.  Even with decreasing anthropogenic NOy it plausible that future increases in some GHGs, e.g., CH4, could lead also to widespread increases in ozone concentrations in the future, thus exacerbating CUO increases under a projected wetter climate in Eastern U.S.

The text here describes the spatiotemporal variability of CUO during 2018-2023, driven by various factors such as the changing emissions, land cover types, environmental and vegetation conditions. Many of these factors are intrinsically interconnected (see Section 1) and the spatially and temporally varying land-atmosphere coupling strengths are discussed in Sections 2.2.2 and 3.3.1. In terms of model uncertainty, the first two case studies offer insights into the SM impacts on regional $O_3$ and Nr and how one may improve models in these aspects. In this paper, we also indicate the model's incapability of accurately representing the impact of stratospheric intrusions on (near-)surface $O_3$ and the impact of omitting spatial variability in $CO_2$ forcing on photosynthesis and $O_3$ uptake. In addition, the estimated overall temporal changes in CUO from this work were discussed together with conclusions in Clifton et al. (2020), which were reached from one global climate model running with the RCP8.5 (now specified in paper). In Clifton et al. (2020), the changes in drought conditions from past to future as well as their associated uncertainty were not explicitly discussed. However, Cook et al. (2020) presented the projected SM, runoff and precipitation changes based on experiments with CMIP5/RCP8.5 and CMIP6/multiple SSPs. Those multimodel based results suggest drier soil conditions in future in many of the eastern US regions that could contribute to the overall decreasing CUO trends. Results in Cook et al. (2020) have been cited in Intergovernmental Panel on Climate Change (2021) and now also included in our discussions.

Cook, B. I., Mankin, J. S., Marvel, K., Williams, A. P., Smerdon, J. E., and Anchukaitis, K. J.: Twenty-first century drought projections in the CMIP6 forcing scenarios, Earth's Future, 8, e2019EF001461, https://doi.org/10.1029/2019EF00146, 2020.

Intergovernmental Panel on Climate Change: the Sixth Assessment Report, Summary for Policymakers, https://www.ipcc.ch/report/ar6/wg1, 2021.

Lines 447-449: This seems a relatively small impact of the SMAP SM DA on surface ozone bias and RMSE, relative to other much larger controlling factors on ozone formation and loss processes.
We now more clearly describe the DA impacts, which are non-trivial. "*Due to increased upwind pollution contributions whereas weakened local emissions and production, both enhancements and reductions by up to ~4 ppbv in daytime surface $O_3$ levels (not shown in figures) are found in the New England region (40.5–43°N, 70–74°W). Across the New England region, WRF-Chem daytime surface $O_3$ performance for 14 July was improved in 31 out of 50 of the model grids where AQS data were available, with the largest improvement of ~1.8 ppbv*".

There are many factors controlling models' $O_3$ performance. Improving one or more processes via DA often does not improve (or even degrade) $O_3$ performance due to the impacts of other error sources. And in such cases, free-running systems perform better for wrong reasons. It is encouraging to find that, in this case, $O_3$ performance was overall improved via the DA. It is unclear what exact "other controlling factors on $O_3$ formation and loss" this reviewer referred to and how large this reviewer estimates their impacts may be for this period/area.

Lines 465-466:  This does seems more important locally (e.g., Northern Virginia), and would be much easier to see if paired ozone spatial bias plots (against AQS) were provided for both the Model no-DA vs. Model DA, instead of having to qualitatively compare the surface AQS obs against contour plots in Figure 14.  Ultimately, I am concerned of the statistical significance of these ozone changes, and think some quantification of the significance is needed here.
A set of figures (Fig. S18) has been added to help better understand the DA impact on the modeled daytime $O_3$ interannual variability. Specific figure contents are: 1) the differences between Figs. 14i and 14h; 2) the *p* values of Student's *t*-tests that compare no-DA and DA daytime surface $O_3$ in July 2022 and July 2018 in all model grids; and 3) scatterplots of the modeled (no-DA and DA cases) vs. AQS daytime surface $O_3$ interannual differences in/near Northern Virginia. The added information helps to demonstrate that SM DA impacts on the modeled $O_3$ fields can vary year-by-year, due to many factors such as observation availability, the performance of SM in the no-DA case, and land-atmosphere coupling strength. In this paragraph, we now explicitly note that Northern Virginia is one of the subregions where SM DA impacts on the modeled daytime $O_3$ are strong for both July 2022 and July 2018.

Lines 484-508:  In this section 3.2, a provided map of irrigated vs. non-irrigated lands is necessary to interpret the changes in Figure 15.  Also, very difficult to interpret the noisy signal of Nr deposition, and as above comment, the significance of these changes are strongly in question for relevance and understanding.  Suggest a statistical significance test is included on these changes, otherwise, the results presented here are very questionable.

Noah-MP's irrigation fraction input based on Salmon et al. (2015) is now shown in Fig. 1c. The soil type map (previous Fig. 1c) is now Fig. S1. Student's *t*-tests comparing daily full/reduced/no irrigation Nr deposition fluxes in all model grids suggest that the most meaningful irrigation impacts (where $p<0.05$, areas marked in green in Fig. 15) on Nr deposition are in/near the irrigated lands in the Carolinas.

Lines 520-522:  I do not follow this argument, ¼-1/3 as large?  This could stem from writing issues here, but very difficult to take anything from this argument scientifically.
Changed to: *"...only ¼–1/3 of its impact on free tropospheric $O_3$"*. See the added Fig. S21 for surface-level stratospheric $O_3$ tracer results. Stratospheric impact on $O_3$ aloft based on the WACCM stratospheric $O_3$ tracer is mentioned in the previous sentence. We recommend using the horizontal and vertical gradients of these $O_3$ tracer fields to help understand the stratospheric $O_3$ impacts instead of interpreting their absolute values as the stratospheric contributions to $O_3$.

Lines 529-540:  These are very weak scientific arguments, and is only very cursory here with the ozone profiles in Fig. 17b and WRF-Chem/AQS spatial maps of ozone concentrations in Figure 17c-j.  There is really no evidence provided here really isolating the elevated ozone with fire plume transport vs. other sources of extra regional ozone and precursor transport, when simply using clean (unrealistic) vs. base simulations in Figure 18.  Indeed, these interactions are known to be very complex regarding ozone concentrations.  More evidence and potentially source apportionment or sensitivity studies (e.g., simply fires on vs. fires off) would be needed to associate these areas with fire plumes vs. other sources of important precursors, and the related enhancements in daytime surface ozone formation.
The "clean BC" simulation is designed to help indicate upwind source (both fire and non-fire) impacts on the study area. Aside from stratospheric intrusions and Canadian fires, cross-state transport of pollution is a policy-relevant topic lately catching lots of attention: e.g., https://www.npr.org/2024/06/27/nx-s1-4996428/supreme-court-good-neighbor-plan , and many other media sources. A sentence has been added to make this point clearer. The fact that different scales of transport (e.g., trans-Pacific, stratospheric intrusions, and interstate) can be dynamically and chemically coupled to impact the western US $O_3$ was demonstrated in previous work (Huang et al., 2013).

An additional simulation "Sen" using chemical boundary conditions (BCs) from WACCM with an alternative fire emission input has been conducted. Fire emission has been identified as one of the most important configurations in global wildfire modeling. There is no standard fire-off NCAR/WACCM product for use as chemical BCs, which would also represent unrealistic conditions for the study period anyway.

The base and two BC sensitivity simulations as well as WACCM stratospheric $O_3$ tracers (see the added daily maps in Fig. S21) together help determine the impacts of stratospheric intrusions and transported Canadian fire (and other) plumes on surface $O_3$ during 13-16 June 2023.

Huang, M., Bowman, K. W., Carmichael, G. R., Pierce, R. B., Worden, H. M., Luo, M., Cooper, O. R., Pollack, I. B., Ryerson, T. B., and Brown, S. S.: Impact of Southern California anthropogenic emissions on ozone pollution in the mountain states: Model analysis and

observational evidence from space, J. Geophys. Res. Atmos., 118, 12,784–12,803, https://doi.org/10.1002/2013JD020205, 2013.

Lines 538-539: Would suggest adding more recent literature on the importance of fires, N deposition, and implications for downwind ecosystems. This is a growing field of importance. https://doi.org/10.1016/j.scitotenv.2022.156130; https://library.wmo.int/records/item/62090-no-3-september-2023; see Pages 7-8
Fires radiative/ecosystem impacts is indeed a growing field of importance. We cited the Koplitz et al. paper published in 2021 which described an earlier CMAQ-based study on this topic. A more recent HTAP3-Fires multimodel experiment paper covering this topic (Whaley et al., 2024, submitted to GMD in July 2024) is now also cited.

**Summary and Suggested Future Directions**

Lines 580-595: I find these arguments significantly broad and not well supported by the presented results in this paper. From what is presented, it is very difficult to determine, where this WRF-Chem configuration performed "remarkably" better than other platforms. Better identification, quantitative comparisons, and examples of improved results are needed. I assume much of this comment is pertaining to the inclusion of Land DA for SM and different simple case studies such as irrigation switches and turning off chemical LBCs, i.e., Clean scenario (Section 3.1-3.3). However, as presented, it is rather cursory arguments, which are not fully apparent how much better this system is able to represent the interactions of Nr and ozone formation.
Operational air quality forecasting systems can undergo multiple times of upgrades within years. Technical notes and peer-reviewed papers have been produced to document these upgrades and their impacts on model performance. There are such papers published in or currently under review for GMD, some of which are cited in Section 1. In such papers, old and new versions of models were usually run for a short period of time and the model results were compared with observations to demonstrate the effectiveness of the model upgrades.

This study serves different purposes, as noted in Section 1. To help determine $O_3$ spatiotemporal variability and the sources and processes controlling it during a multi-year period (including surface-atmosphere interactions which have growing importance and are understudied), high-resolution simulations with relatively consistent configurations and stable performance throughout the study periods are needed. As indicated in various sections of the paper and reiterated here, referring to AQS data, the model's $O_3$ performance is stable and overall better than what's reported in many previous studies (see some examples in Section 1, and language here has been adjusted). Certainly, many factors can contribute to successful model simulations, but through case studies, we highlight the importance of several of them and the needs to further investigate them. We do not deemphasize the importance of other factors controlling the models' $O_3$ performance, and in fact they were carefully considered (benefiting from test runs for short periods) as the baseline simulation was being set up.

The model simulation described here is already extended to 2024 (the TEMPO era), running on a routine basis. The study has connections with various communities, as well as implications for

updating other models in the aspects being highlighted. This point is now explicitly made in this section along with extended uncertainty discussions.

**Technical Corrections:**

***Detailed grammatical errors and suggestions only shown here for Abstract and Introduction sections\*\*\****

**Abstract**

Lines 15-20:  Grammatical error.  Run-on sentence, and needs revision.
Revised.

Line 20:  Grammatical error.  This statement "compared with and related to" is redundant.
 "Compare with" has been removed.

Lines 23-24:  Grammatical error.  Remove comma.
Done.

**1 Background, motivation, and goals**

Lines 42-44:  Grammatical error.  The sentence structure is very awkward, and needs revision.
Revised.

Line 47:  Grammatical error.   Change "O3 via the aerosol radiative" to "O3 via aerosol radiative". Lines 47-48:  Grammatical error.  Remove "the" in "via the aerosol radiative effects".
Done.

Lines 48-54:  Grammatical error.   Run-on sentence, and needs revision.
Revised.

Line 55:  Suggest changing  "would be" to "is".
Done.

Lines 58-59:  Grammatical error.  Awkward sentence structure, and cannot understand the connection the author is making with  "…and carbon dioxide (CO2) concentration as well as plants' physiological conditions."
Changed to: "*closely interact with multiple other interconnected environmental stressors (e.g., temperature, humidity, precipitation, soil moisture, SM, and carbon dioxide, $CO_2$) and plants' physiological conditions*".

Lines 62-64:  Grammatical errors and inappropriate verbiage.  "…continue to decrease there due..", "…for studies on Nr and O3, attention should…", and "imported".
This sentence has been broken down and reworded.

Line 68:  Grammatical error.  Need comma, "…and the estimated background O3, as well as…"

*This sentence has been broken down and reworded.*

Lines 74-75:  Grammatical error. Run-on sentence, and needs revision.
*This sentence has been broken down.*

Line 76:  Grammatical error.  Awkward sentence structure.  "…limits the capability of understanding air quality there and evaluating…"
*Changed to: "limits our capability of understanding air quality there and evaluating..."*

Line 78:  Awkward verbiage.  Suggest changing "is anticipated to"  to "will".
*Changed to "can".*

Lines 80-85:  Grammatical error.  Run-on sentence, and needs revision.
*Revised.*

Lines 88-97:  This is too long for a bulleted list, particularly difficult to read in bullet 3).  Suggest separating it out of the paragraph and shortening into more bullets to make easier to read and understand.
*The three bullets correspond to Sections 3.1, 3.2, and 3.3 of the results section. We have changed the structure of the sentences/phrases related to bullet 3).*

**Results**

Lines 318-339:  I find this section needs significant writing improvements, as discussed above.  Also, it would be best to break this paragraph up into multiple paragraphs.
*See response to your earlier comment on these lines.*

Lines 341-365:  Writing needs significant improvement and needs multiple paragraphs.
*See response to your earlier comment on these lines.*

Lines 381-431:  Writing needs significant improvement and sentence structure needs substanitial improvement.  Currently it is difficult to follow the arguments.
Note:  Similar writing improvements are needed through the remaining results section.
*See response to your earlier comment on this paragraph. These lines have been broken down and revised.*

**Summary and Suggested Future Directions**

Technically, this section also needs similar significant writing improvements and is very cumbersome to read.  Highly recommend thorough proof-reading in revised manuscript.
*This section has been revised according to your earlier comments and proofread.*

**Response to RC2**

The MS (egusphere-2024-484) by Huang conduced model simulation of Reactive nitrogen in and around the northeastern and Mid-Atlantic US, and analyzed its influence on O3 and plant, it shows a lot of model simulation work by considering different model setups and also analyzing so many components, which follows Huang's previous studies as list in references. However, the performance of model simulation, especially for dry/wet deposition and the influence of O3 on plants should be furthered carefully evaluated, which can be potential large uncertainty for this study.

Thanks for the summary. Please see below for the added model evaluation work and discussions on sources of model uncertainty.

other comments as:

Line 111-113, "Noah-MP's CO2 forcing for 2018, 2019, 2020, 2022, 2023's warm seasons were set to 410, 412, 415, 420, and 423 ppmv, respectively, based on measurements at the Mauna Loa Observatory and its nearby Maunakea Observatories for part of 2023", why this study choose GHG background values of Mauna Loa as the CO2 forcing for the urban area, which can have much higher CO2 concentration as >450 ppm, and affect photosynthesis of plants.

The $CO_2$ forcing for the Noah-MP land surface model is typically set as a constant value and therefore including year-to-year changes in that forcing is already an advance.

The increases in $CO_2$ are seen at the Mauna Loa Observatory and across the globe at similar speeds of approximately 2-3 ppmv year$^{-1}$ for recent years, according to observations from the NOAA GML network, satellites, and model/analysis fields (e.g., https://nasaviz.gsfc.nasa.gov/5194; https://gml.noaa.gov/ccgg/trends/gl_gr.html; https://gml.noaa.gov/webdata/ccgg/CT2022/CT2022.global_AGR.pdf). We recognize the different magnitudes and seasonal variability of rural and urban $CO_2$ (Fig. 11 in Karion et al., 2020), which in some years present anomalies due to COVID (Weir et al., 2021), and these were not accounted for in our configurations. However, the impact of ignoring these differences (tens of ppmv) on the photosynthesis-based dry deposition estimates is likely to be very small according to independent global model sensitivity analysis (e.g., Fig. 12 in Sun et al., 2022; Fig. 7 in Silva et al., 2023), and is worth future investigations with finer-resolution models. The need to develop high-quality, spatially and temporally varying $CO_2$ forcings for Noah-MP, especially in their longer-period simulations, was brought up at a recent Noah-MP Users' International and similar occasions. This is now also mentioned in Section 4.

Karion, A., Callahan, W., Stock, M., Prinzivalli, S., Verhulst, K. R., Kim, J., Salameh, P. K., Lopez-Coto, I., and Whetstone, J.: Greenhouse gas observations from the Northeast Corridor tower network, Earth Syst. Sci. Data, 12, 699–717, https://doi.org/10.5194/essd-12-699-2020, 2020.

Silva, S. J., Burrows, S. M., Calvin, K., Cameron-Smith, P. J., Shi, X., and Zhou, T.: Contrasting the biophysical and radiative effects of rising $CO_2$ concentrations on ozone dry deposition fluxes, J. Geophys. Res. Atmos., 128, e2022JD037668, https://doi.org/10.1029/2022JD037668, 2023.

Sun, S., Tai, A. P. K., Yung, D. H. Y., Wong, A. Y. H., Ducker, J. A., and Holmes, C. D.: Influence of plant ecophysiology on ozone dry deposition: comparing between multiplicative and photosynthesis-based dry deposition schemes and their responses to rising $CO_2$ level, Biogeosciences, 19, 1753–1776, https://doi.org/10.5194/bg-19-1753-2022, 2022.

Weir, B., et al.: Regional impacts of COVID-19 on carbon dioxide detected worldwide from space, Sci. Adv., 7, eabf9415, https://doi.org/10.1126/sciadv.abf9415, 2021.

Line 155, "missions Database for Global Atmospheric Research version 5 based on the Community Emissions Data", please illustrate what the spanning years for EDGAR v5.0 for these pollution species.
Added "*for the years after 2015*".

Line 165-174, the authors mentioned the annual variations of different pollution species, it's much better to illustrated them with time series figure than worlds.
Agreed. That information is indicated in Fig. 6a.

Section 2.2.3 ground-based observations, why the site based PM2.5 PM10 the components SO42- NO3- NH4+ were not compared with model simulations, which can support your model's performance regarding atmospheric chemical reaction, dry/wet deposition and pollution emissions.
The following model evaluation work has been added, along with discussions:
  1) Wet deposition fluxes of $SO_4$, $NO_3$, and $NH_4$, as well as precipitation, evaluated with NADP/NTN data;
  2) Surface $SO_4$, $NH_4$, and $NO_3$ concentrations evaluated with CASTNET (remote/rural) and AQS (urban/suburban) observations;
  3) Surface $HNO_3$ concentrations evaluated with CASTNET observations; and
  4) Surface $NH_3$ concentrations evaluated with NADP/AMoN data.
Additionally, literature on surface speciated aerosol trends based on the IMPROVE observations (Hand et al., 2024) is now also cited in the Supplement.

Hand, J. L., Prenni, A. J., and Schichtel, B. A.: Trends in seasonal mean speciated aerosol composition in remote areas of the United States from 2000 through 2021, J. Geophys. Res. Atmos., 129, e2023JD039902, https://doi. org/10.1029/2023JD039902, 2024.

Section 2.3.2 I am still wondering whether the plant models in your study can well represent the harmful O3 effect on stomate, where the parameters and plant model structure can largely affect your evaluation.
As introduced in Section 2.1, $O_3$ vegetative impacts were dynamically modeled by applying two separate factors $F_{p,O3}$ and $F_{c,O3}$ (which are linearly related to CUO) to photosynthesis and stomatal conductance rates, and an $O_3$ flux threshold to account the ability of plants to detoxify $O_3$ was applied.

Key sources of uncertainty include: 1) methods to calculate stomatal conductance/resistance, including model structures; 2) slopes and intercepts used to estimate $F_{p,O3}$ and $F_{c,O3}$ for a limited number of plant types; and 3) the $O_3$ flux threshold used to account for the ability of plants to

detoxify $O_3$. 1) has been discussed in depth in many previous studies including Huang et al. (2022), Sun et al. (2022) and references therein. The Ball-Berry type of approaches have shown advantages over multiplicative and Medlyn based approaches. Some of the inputs of the stomatal models such as SM could be improved through data assimilation, as highlighted in Huang et al. (2022). Improving the model's $CO_2$ forcings and assimilating other datasets are encouraged (Section 4), according to both reviewers' comments. 2) and 3) were taken from literature with evidence based on measurements, which can certainly be uncertain for this case, and we note in Section 4 that evaluating and improving these parameters for more types of plants at various growth stages in future is encouraged.

Sun, S., Tai, A. P. K., Yung, D. H. Y., Wong, A. Y. H., Ducker, J. A., and Holmes, C. D.: Influence of plant ecophysiology on ozone dry deposition: comparing between multiplicative and photosynthesis-based dry deposition schemes and their responses to rising $CO_2$ level, Biogeosciences, 19, 1753–1776, https://doi.org/10.5194/bg-19-1753-2022, 2022.

Section 3.1 usually the model simulated results should first compare with observations (not all species, depends on what observations the authors have as illustrated in method section) to verify the performance of model. It's easy to run the model and analyzed model simulations, but it's hard to tell us whether the simulations from your model parameter and emission setup were reliable. Here on line 407, I just notice your comparison with surface O3, and I am not that confident your model can well simulate the spatial-temporal variations of O3 changes, as you only displayed the averages of a period, instead of hourly observations, with considerable bias. Have you considered the impact of stratospheric intrusion on ozone enhancement in the lower troposphere with upper O3 boundary condition scheme?

Section 3.2 focuses on interannual differences in $O_3$. In case study #3, we added a timeseries plot of the domain-mean observed and WRF-Chem hourly surface $O_3$ during 13-16 June 2023 at AQS sites (current Fig. 17c). This plot is now discussed together with $O_3$ timeseries for MJJ 2023 (Fig. S24) and the model performance during other fire events in MJJ 2023 (below).

[Figure]

The challenge for regional chemistry models, especially those systems without accurate dynamic upper chemical boundary conditions, to well capture the impacts of stratospheric intrusions on $O_3$ enhancements had been mentioned here in Section 3.2 as well as our previous studies, and now also explicitly in case study #3. This challenge contributes to the slight negative biases in

WRF-Chem daytime O$_3$ for this case, but the model still did an excellent job in reproducing the observed hourly O$_3$. An important message of this paper is that proper updates on WRF-Chem related to land and land-atmosphere interactions are transferrable to other regional models, including those running with dynamic upper chemical boundary conditions (now mentioned in both Section 3.3.3 and Section 4).

Section 3.2 irrigation approaches, on line 486, the "Ozone perturbs gross primary productivity more strongly (up to 20–30%) than transpiration", as I mentioned above, whether there are observation-based study that displayed similar results? because the plants model can not well simulate the feedback between O3 and plant. To me The GPP decreased by 20-30% only caused by O3 is not reliable, see the situations in China and India, large O3 concentrations occurred in summer, but the influence on crop production did not change too much. From my experience, even the influence of O3 on plants have not been well investigated by field observations, how can it be well represented by model equation and structure?

Please see our response to your earlier comment on Section 2.3.2. Also, the modeled GPP from the baseline simulation (including O$_3$ impacts) was compared with the MODIS Terra/Aqua 8-day GPP product (also known to be uncertain despite its wide usage, especially for locations with frequent cloud cover and high GPP) for June 18-25, 2022. Both the model and MODIS indicate that GPP was <0.04 kgC m$^{-2}$ over dry croplands and 0.10-0.12 kgC m$^{-2}$ over humid forest regions.

A key point from this paper is that the O$_3$ impacts on surface fluxes and vegetation are sensitive to various environmental factors. We are not sure under which conditions "in China and India, large O$_3$ concentrations occurred in summer, but the influence on crop production did not change too much". This finding may be cited if more information can be provided. Fig. 3 in Lombardozzi et al. (2015) shows 20-year average ozone impacts on GPP across the globe - for some places in the US, Asia, and Africa, these impacts were estimated to be >25%. Some of us are aware of multiple papers on the O$_3$ impacts on various types of ecosystems by TOAR-II vegetation team members such as Pandey et al. (2023) for India as well as a few in-review and in-preparation papers for this special issue. Results from these studies are/may also be informative.

Pandey, D., Sharps, K., Simpson, D., Ramaswami, B., Cremades, R., Booth, N., Jamir, C., Büker, P., Sinha, V., Sinha, B., and Emberson, L. D.: Assessing the costs of ozone pollution in India for wheat producers, consumers, and government food welfare policies, Proc. Natl. Acad. Sci., 120(32), e2207081120, https://doi.org/10.1073/pnas.2207081120, 2023.

---

## Author Response (AR2)

**Please see below our point-by-point response (in blue) to both reviewers' comments (in black). Quoted text from the revised manuscript is *in italic*.**

**The most updated (20 August 2024) US EPA AQS surface measurements are included in this revision. Color-blind friendly color schemes are applied to figures, referring to Crameri et al. (2020, https://doi.org/10.1038/s41467-020-19160-7) and other references in ACP's submission guidelines.**

Report 1 (Referee #3)

Figure 6: I found this figure difficult to follow in relation to the text. Are NH3 (anth+ fire) emissions considered part of the Nr emissions? Additionally, what is the difference in emissions between water and land? Does 'anth NOx emission (water)' refer to shipping? Better notations on the figure would greatly enhance both the figure and the associated discussion.

Thanks for the questions. The definitions of "reactive nitrogen (Nr)" in literature vary. In some cases, it includes $NH_x$, and in the others, it does not. In this article, while oxidized nitrogen is more emphasized, $NH_x$ is also included in reactive nitrogen. See definition of Nr in Section 1. Labels like ".. emission (water)" refer to emissions from the model grids that are classified as water. These are not necessarily the same as shipping emissions, because some of the anth emissions from the shipping sector are assigned to grids overland (e.g., ports and surrounding areas). We added "*water and land model grids are defined in Fig. 1b*" to Fig. 6 caption.

Lines 425-430 and Figure 11: I'm unclear on what is meant by a higher correlation between O3 and the NO2 column compared to the HCHO column. This information doesn't seem directly applicable to identifying NOx-sensitive chemical regimes, which are typically determined by the relationship between P(O3) (O3 production rate) and NOx levels. What are the key takeaways from these correlation results?

Good point. Fig. 11 has been updated, which now indicates the $NO_2$ columns-daytime surface $O_3$ relationship as well as its dependency on column $HCHO/NO_2$ ratio. Relevant sentences in this paragraph and elsewhere are modified to explicitly draw implications from this plot (along with other results discussed in this paragraph) regarding the effects of $NO_x$ changes on $O_3$ in this area, as well as the utility of remote sensing $NO_2$ and HCHO column data in inferring surface $O_3$ variability across the area. Please also see the previous paragraph on satellite $HCHO/NO_2$ as an indicator of chemical regimes.

Report 2 (Referee #1)

Lines 68-79: The studies reported here about ability of regional models to model surface ozone are outdated, and reference DA attempts to improve results dating back to 2007. Thus, I think stating biases of surface O3 up to 20 ppbv is truly not representative of the state-of-science regional CTMs/AQMs that are available today for the U.S. CMAQv5+ for instance can well simulate surface O3 with biases much less than 20 ppbv in the U.S. without significant DA (e.g.,

CMAQ). Some Examples. 1) Offline CMAQv5.3.1, Appel et al. (2021) https://gmd.copernicus.org/articles/14/2867/2021/, reports surface ozone well within +/- 5 ppb. ). 2) Two-Way Coupled WRF-CMAQ with Noah LSM updates: Campbell et al. (2019), https://agupubs.onlinelibrary.wiley.com/doi/full/10.1029/2018MS001422 show improvements in ozone without DA. This should be at least discussed here for state-of-science CTMs in context to fully coupled ESM configurations that also try to model surface ozone (maybe less successfully). Furthermore, its worth noting that when using well constrained, bottom=up emissions for the simulation period (e.g. NEI) and robust chemical mechanisms (and even empirical approaches to dry deposition) offline or online CTMs like CMAQv5+ are shown to recently perform very well for surface ozone in the U.S. (<< +/-5 ppb).

Changed to: "*...large model-observation mismatches in surface O$_3$ of up to tens of ppbv were not well explained or attributed mainly to the models' uncertain/outdated anth emission inputs...*"

We do not agree with this reviewer that ESMs (which also evolve quickly through time) perform less successfully. We point out that the overall model biases reported in some of these suggested papers (and other modeling works) are results of positive and negative biases in different US regions being cancelled out, and Appel et al. (2021, suggested by this reviewer) showed that CMAQ updates degraded the model performance for some US regions/seasons. Additionally, none of these studies include source attribution analysis which is a discussion point in this paragraph. As this reviewer noted, model performance is highly dependent on their inputs and parameterizations, and the same models' performance (including CMAQv5.3.1+ runs at regulatory and operational agencies and in academia) can vary substantially in different applications. This is also supported by Figure S1 of Hogrefe et al. (2023, process level study) that showed big differences in Appel et al. (2021) and AQMEII CMAQv5.3.1 O$_3$ and aerosol fields. Model simulations with the bottom-up NEI (not available for every year) emissions for their base years over the US are supposed to lead to better model performance than for non-NEI years. Such findings are in some of the papers we already cited, which also demonstrate that chemical data assimilation is an effective approach to improve the NEI, especially for non-NEI years. The choice of chemical mechanism may also impact O$_3$ in regional models by several ppbv according to numerous existing sensitivity studies.

Campbell et al. (2019) demonstrate that tuning several sets of static, hard-coded parameters can improve the modeled (with Noah LSM and empirical dry deposition methods) air pollution fields for slightly over 50% of their grids and the model performance in other grids was worsened. Certainly, tuning static parameters is one way to improve models. However, in general, less complex modeling systems with empirical approaches for processes like dry deposition are less suited to evaluate the sensitivities of air pollution states and processes to various climatic factors (e.g., Niyogi and Raman, 1997, and many later studies, on assessing stomatal resistance by different schemes). Evaluating air pollution responses to climate change has become increasingly important to help better understand the Earth systems and their interconnectivity as well as assisting in developing emission control strategies. Less complex modeling systems may be

more computationally efficient, have fewer and more easily identifiable sources of uncertainty, and their pollution fields can be less responsive to the applications of data assimilation that adjust environmental and biophysical conditions. Therefore, models and their configurations should be chosen based on the objectives of research and applications and carefully evaluated.

Hogrefe, C., Bash, J. O., Pleim, J. E., Schwede, D. B., Gilliam, R. C., Foley, K. M., Appel, K. W., and Mathur, R.: An analysis of CMAQ gas-phase dry deposition over North America through grid-scale and land-use-specific diagnostics in the context of AQMEII4, Atmos. Chem. Phys., 23, 8119–8147, https://doi.org/10.5194/acp-23-8119-2023, 2023.

Niyogi, D. S. and Raman, S.: Comparison of Four Different Stomatal Resistance Schemes Using FIFE Observations, J. Appl. Meteorol. Climatol., 36, 903–917, https://doi.org/10.1175/1520-0450(1997)036<0903:COFDSR>2.0.CO;2, 1997.

Lines 167-168: There are many studies on the impacts of background NOx sources when inferring emissions from satellite sources in the literature. This sentence should be revised with adequate citations, e.g., Silvern et al. (2019): https://doi.org/10.5194/acp-19-8863-2019 Qu et al. (2021) https://agupubs.onlinelibrary.wiley.com/doi/full/10.1029/2021GL092783 East et al. (2022) https://acp.copernicus.org/articles/22/15981/2022/acp-22-15981-2022.pdf

As noted in our initial response, the related point of view in some of these suggested papers has been covered in Elguindi et al. (2020) and we think this citation is sufficient for this context. The lightning $NO_x$ biases shown in East et al. (2022) were also recognized by Elguindi et al. (2020) qualitatively and quantified in much earlier studies, such as:

Jourdain, L., Kulawik, S. S., Worden, H. M., Pickering, K. E., Worden, J., and Thompson, A. M.: Lightning $NO_x$ emissions over the USA constrained by TES ozone observations and the GEOS-Chem model, Atmos. Chem. Phys., 10, 107–119, https://doi.org/10.5194/acp-10-107-2010, 2010.

Miyazaki, K., Eskes, H. J., Sudo, K., and Zhang, C.: Global lightning $NO_x$ production estimated by an assimilation of multiple satellite data sets, Atmos. Chem. Phys., 14, 3277–3305, https://doi.org/10.5194/acp-14-3277-2014, 2014.

Lines 181-182: Since the driving meteorological reanalysis dataset and corresponding WRF land/meteorological simulations can strongly control the chemistry and surface-atmosphere exchange processes, I feel that a base evaluation of the WRF output across eastern U.S. domain (against 2D/3D Met observations like METARs/RAOBs/BSRN/PRISM) is lacking here. Particularly when downscaling from relatively coarse 32 km ICs/BCs. I suggest that some basic meteorological evaluation are included at least in the supporting information and discussed in later results section to help qualify the overall Met results and discuss implications for Met biases on AQ and sfc-atm-x processes. This could also prove very useful to discuss impacts of the land/SM DA on the Met performance, not just quantifying the potential influences of land-->weather changes on AQ. For example, does the WRF Met performance also improve when including the land/SM DA? Why or why not?

Thanks for the suggestion. The multi-year model precipitation and temperature performance is already indicated in several figures. Our previous SM DA case studies include detailed met evaluation. In this paper's case study, evident improvements in the modeled air temperature due to SM DA are now presented (Section 3.3.1/Figs. 14 and S18), which contributed to the improved surface $O_3$ performance.

Lines 185-186: This sentence is unclear. Are the authors saying that other chemical reanalysis products are more accurate than WACCM, or vice versa? Also, references/citations are needed here to support this statement.

Changed to: "*are likely to be more accurate*".

While there are many references on chemical reanalysis products, the conclusions therein cannot be directly applied to this study. We did not compare the chemical boundary models used and chemical reanalysis products for our study period. However, model sensitivities to chemical boundary conditions have already been presented in a case study (Section 3.3.3).

Line 188: Awkward wording to use "hiked by". Please revise.

Changed to "rose".

Lines 272-273: This relates to my earlier comment on adding Met evaluation with independent observations, not just quantify impacts.

Please see our response to your earlier comment on met evaluation. "*Impacts*" refer to the resulting changes in model fields and their accuracy.

Line 353: This sentence does not make sense. Do you mean "greatly resemble one another"?

Changed as suggested.

Line 418: "Manifests" is a very awkward writing in this sentence.

Changed to "indicates".

Line 426: Sentence is incomplete, please revise to "resemble one another".

Changed as suggested.

Line 427: This part of the sentence is also poor grammar/sentence structure,. please revise.

This sentence has been broken down into two sentences.

Lines 432-434: This is again poor sentence structure, run-on, missing appropriate commas.

Comma added.

Lines 438-444: This effect is spatiotemporally variable in the U.S., with some increases in daytime ozone due to COVID-19 induced emissions changes. See Figures 6-7 in: https://doi.org/10.1016/j.atmosenv.2021.118713.

These are domain-wide results. Spatial patterns of daytime surface $O_3$ fields are shown in Fig. 10 of this article.

Line 448: Please see earlier comment. I don't think that such large bias/error of ~ 20 ppb is common amongst state-of-science regional AQMs/CTMs. Please revise here and above.

Changed to "*tens of ppbv*", also accounting for RMSEs reported in Appel et al. (2021) for this region/warm seasons.

Lines 470-471: I don't agree with this argument based on Figure 12b. This simply shows the magnitude of Nr deposition to different vegetation and water. However, the relative ecosystem impacts plotted in some way (as a function of impact on herbaceous plants, lichen species, algae blooms/acidification/anoxic conditions, etc.) may be a whole different thing in regards to impacts. Suggest revising.

Fig. 12b was cited at the correct location, following "*The potential impacts of Nr deposition are strongest and weakest on croplands and water, respectively*". Figs. 8b-c and S15 are now cited immediately following the previous sentence, which together indicate the year-to-year changes in the speciated deposition fluxes and their respective contributions to the total fluxes.

Lines 488-509: If the observed large surface ozone of about 30 ppb was due largely due to the impacts of frontal passage, precipitation and soil moisture changes, then the improvement due to SSM DA of ~ 2 ppb is only about a 5% of this change. Likely the largest primary contribution to this drop is the direct weather effects resulting cleaner airmass and lower temperatures behind the frontal passage, not necessarily SSM responses and secondary weather feedbacks. I think this limitation should be better discussed.

A ~2 ppbv change in ambient $O_3$ concentration is non-trivial considering the economic cost of air pollution reduction. A ~2 ppbv reduction in the modeled $O_3$ bias is not small considering the many factors that can impact the $O_3$ performance. This is in fact better than/comparable to the improvements in $O_3$ for the New England region due to updating Noah LSM parameters of a WRF/CMAQ system shown in Campbell et al. (2019, suggested earlier by this reviewer). Please note that all the impacts discussed in these lines (not the 30 ppbv drop) must be attributed to SM DA, as they were determined from the no-DA and DA cases.

We agree that investigating SM DA impacts on $O_3$ and other variables across three dimensions under various weather conditions is an interesting direction - please see our previous studies during other field campaigns conducted in the US and Asia, several of which have been cited in this paper. Also, in the following paragraphs and Figs. 14 and S18, we highlight larger SM DA

impacts (>4 ppbv) on the model's surface $O_3$ performance in other eastern US regions on interannual timescale.

Lines 591-592: Also see https://www.sciencedirect.com/science/article/pii/S0048969722032272 and https://library.wmo.int/records/item/62090-no-3-september-2023 (pp 7-8).

The approach in Campbell et al. (2022) is similar to that in an earlier paper we already cited. One of its NOAA authors' affiliation is written wrong. As this suggested paper appears that it was not carefully proofread and very likely not internally reviewed/approved by all relevant NOAA offices prior to its submission (a NOAA requirement), we do not cite it.

Line 627: This sentence is incomplete ..."caused biomass/crop yield losses by a few percent". In which direction?

Loss means reduction.

643-644: I suggest revising this based on my earlier comments, as such large ozone biases/error quoted in this paper is not reflective of state-of-the-science AQMs.

See our responses to your earlier comments. Corresponding sentences in other places describing the varying model performance for this region/warm seasons have been changed to "*tens of ppbv*".

---

## Author Response (AR3)

**ACP's continued efforts on this paper are greatly appreciated. Please see below the author responses (in blue) to Editor's suggestions (in black). Quoted text from the revised manuscript is *in italic*.**

Editor's suggestion (based on Referee #3's earlier comment on L425-430 and Figure 11): the explanation of the above query is insufficient. The authors should add lines on the below point: "In Figure 11, the high FNR in the region where the correlation between ozone and NOx is high indicates NOx-sensitive regime."

Changed to: "*Fig. 11 indicates the connection between early afternoon (19 UTC) $NO_2$ columns and daytime surface $O_3$ as well as the dependency of this connection on column $HCHO/NO_2$ ratios. Larger-than-two $HCHO/NO_2$ values dominate the study region where the overall surface $O_3$-$NO_2$ column spatial correlation is high (r=0.54). Daytime surface $O_3$ concentrations exhibit the most robust spatial correlation with early afternoon $NO_2$ columns in 2020 (r=0.62, versus 0.47–0.56 for other years), when the domain-wide median and mean $HCHO/NO_2$ ratios are larger than the other years' by at least 0.5. These model results suggest that $NO_x$-sensitive or transitional regimes dominate this region during 2018–2023 and point to a potential of inferring surface $O_3$ variability across this area from high-quality remote sensing $NO_2$ and HCHO column data*".*

Editor's suggestion (based on Referee #1's earlier comment on Lines 181-182): Please add discussions and references of previous papers on SM-DA case studies where other met parameters are evaluated.

Referee #1 questioned about NARR's data quality and its impact on WRF's met performance. To address this, two references have been added to this paragraph: "*Daily reinitialized atmospheric initial conditions (ICs) and boundary conditions (BCs) were downscaled from the 3-hourly, 32 km North American Regional Reanalysis (NARR) dataset, which overall well represents the observed daily variability in apparent temperature for the eastern US (e.g., Ibebuchi et al., 2024). Huang et al. (2017b) showed that, initializing WRF with the North American Mesoscale Forecast System (6-hourly, 12 km)'s atmospheric fields instead of NARR's did not result in significant changes in WRF-simulated surface air temperature fields over the southeastern US*". Ibebuchi et al. (2024) also found that NARR's apparent temperature performance is not as good as that of ERA5 (at 0.25°×0.25° horizontal resolution). In this paper, Fig. S19 shows overall consistent year-by-year variability in WRF and ERA5 surface air temperature.

The impacts of SM DA on non-temperature 2D/3D met fields (e.g., humidity, winds, precipitation) can be found in Huang et al. (2021, 2022, over the southeastern US, WRF initialized with NARR) and Huang et al. (2018, over Asia, WRF initialized with FNL). Results in Huang et al. (2022) indicate that % improvements in near-surface and free tropospheric air temperature due to SM DA are greater than those in humidity for Noah-MP based cases. A sentence has been added to Fig. S18 caption: "*The improvements in other key meteorological*

*fields due to the DA, which may be relatively smaller than that in air temperature according to previous studies (e.g., Huang et al., 2021, 2022), also impacted the model's O$_3$ performance.*".

Huang, M., Crawford, J. H., Diskin, G. S., Santanello, J. A., Kumar, S. V., Pusede, S. E., Parrington, M., and Carmichael, G. R.: Modeling regional pollution transport events during KORUS-AQ: Progress and challenges in improving representation of land-atmosphere feedbacks, J. Geophys. Res.-Atmos., 123, 10732–10756, https://doi.org/10.1029/2018JD028554, 2018.

Ibebuchi, C. C., Lee, C. C., Silva, A., and Sheridan, S. C.: Evaluating apparent temperature in the contiguous United States from four reanalysis products using Artificial Neural Networks, J. Geophys. Res.-Machine Learning and Computation, 1, e2023JH000102, https://doi.org/10.1029/2023JH000102, 2024.

Editor's suggestion (based on Referee #1's earlier comment on Lines 488-509): As suggested by the referee a line should be added in the manuscript on, 'Meteorology can also contribute partially to the observed enhancement in the surface ozone'.
This point is now more clearly written at the beginning of Section 3.3.1 "*Satellite (i.e., GPM, SMAP, and TROPOMI) and in situ observations collected at/round Harvard Forest and the CRN-Millbrook site during the SMAPVEX22 campaign were analyzed along with WRF-Chem results during a precipitating event associated with a frontal passage that occurred from late 13 July to early 14 July 2022. This event caused sharp increases in SSM around 14 July in Massachusetts (by >0.06 m$^3$ m$^{-3}$) and parts of the eastern New York (by ~0.02 m$^3$ m$^{-3}$), as well as drastic changes in air temperature (by up to ~5 K decreases at the surface) and other meteorological fields. These changes in SSM and meteorological conditions contributed to the abrupt O$_3$ reductions of up to 30 ppbv (Figs. 13a and S17)*".

Please note that SM DA has feedback on frontal passage characteristics, which impact both local and upwind O$_3$ (not always clean as this referee suggested). For example: "*The enhancements in soil wetness resulted in altered precipitation characteristics, a bit cooler surface soil/air, thinner atmospheric boundary layer, suppressed biogenic VOC and soil NO$_y$ emissions as well as O$_3$ formation while deposition accelerated….*".

---

## Author Response (AR4)

Editor's comment: The authors have included suggestions given by two reviewers. The revised manuscript reads well and can be published after technical corrections. The manuscript should be arranged as per ACP guidelines. https://www.atmospheric-chemistry-and-physics.net/policies/guidelines_for_authors.html
Abstract should contain 250 words.

Response: The revised abstract contains 250 words. Minor language edits are indicated in the attached file.

[revised manuscript text omitted]